# Utilising the Social Return on Investment (SROI) Framework to Gauge Social Value in the Fast Forward Program

**Jioji Ravulo** [1,*], **Shannon Said** [1], **Jim Micsko** [2] **and Gayl Purchase** [2]

1   School of Health and Society, Faculty of Social Sciences, University of Wollongong, Wollongong, NSW 2522, Australia; ssaid@uow.edu.au
2   Widening Participation Engagement Marketing, Office of Marketing and Communication, Western Sydney University, Penrith, NSW 2751, Australia; j.micsko@westernsydney.edu.au (J.M.); G.Purchase@westernsydney.edu.au (G.P.)
*   Correspondence: jioji@uow.edu.au; Tel.: +612-8763-6009

**Abstract:** A market paradigm shift towards a 'knowledge-based economy' means Australia is moving towards a major skills crisis whereby the workforce will lack skills attainable from higher education. Moreover, those from low socio-economic backgrounds, and who are confronted with disadvantage, still face challenges in gaining entry to university. The Fast Forward Program (FFP) aims to increase attainment of higher education for greater western Sydney high school students in years 9–12, with a focus on dismantling the social barriers preventing attainment. To achieve this aim, the program hosts a range of student and parent in-school workshops and on-campus visits. To capture the social impact of the program for all participants, the social return on investment (SROI) methodology was implemented. The SROI ratio is represented as a return in dollar value for every dollar invested; due to the success of the program, the investment represented $5.73 for every $1 spent. The key findings indicated that students and parents gained a deeper familiarity and understanding of university which, in turn, created a deeper confidence and motivation for students to enter higher education. Additionally, participants reported being able to better use their time to cater for study, and were more comfortable about going onto a university campus.

**Keywords:** widening participation; low socio-economic background students; higher education; high-school students; social barriers

## 1. Introduction

In 2009, the Australian Government created a series of objectives for increased participation in and attainment of higher education. These goals seek to have 40% of individuals aged 25–34 with a university qualification by 2025, and increase participation of those from low socio-economic backgrounds to 20% by 2020 [1]. From 2010, there has been a heightened effort to encourage students from 'non-traditional' or traditionally under-represented backgrounds to engage with higher education through the Higher Education Participation and Partnership Program, or HEPPP [1]. In order to fulfil these objectives, universities have been engaging with primary and high school communities across greater western Sydney encouraging participation especially among those who do not typically consider university a viable option for their future. Within this context, non-traditional students are those who are typically the first in their family to attend university, from migrant and/or from non-English speaking backgrounds.

This article considers the role of the Fast Forward Program (FFP), run by the Office of Marketing and Communication, at Western Sydney University (WSU), which has multiple campuses located

across greater Western Sydney. FFP seeks to encourage students from years 9 to 12 to aspire to continue their studies through university and other tertiary study, through a range of student and parent in-school workshops and on-campus visits, which feature professional and academic staff sharing their experiences in order to bolster confidence for young people to aspire towards higher education and training. This article considers the social impact of these programs—that is, how perceptions and aspirations of attending university have changed—and how they affect different stakeholders, namely, students, parents, FFP project officers, and academic support staff of Western Sydney University (WSU), discussing this impact through a social return on investment (SROI) methodology. This methodology highlights the impact of the project upon these stakeholder groups through both narrative (qualitative) and quantitative measures.

*The Australian Higher Education System*

The Australian higher education system allows students to apply to the university of their choice once they have completed their higher school certificate or equivalent—different states have different names for the same qualification (for example, the Victorian Certificate of Education in Victoria). Students' scores are determined by their Australian Tertiary Admissions Ranking or ATAR, with specific courses having a specific ATAR score required for entry. ATARs are determined by a combination of in-class assessments completed throughout the final year of high school study, and performance in a range of examinations throughout the year. Students apply in the year prior to their intended commencement of study (usually February/March of the subsequent year) for the courses they wish to study via the University Admissions Centre or UAC [2], and they are offered a place at the university of their choice on the basis of how they have performed against other students who have also applied for the course. Some courses have extra or other entrance requirements, such as medicine, which are determined by each university. Other entry pathways are also possible, such as applying directly to the university [2] or studying at Technical and Further Education centres [3] or other colleges that offer pre-university level certificates, which can be used to transition into university courses. Tuition fees are typically offset by the Australian Federal Government's Higher Education Contribution Scheme – Higher Education Loan Program (HECS-HELP) scheme, where domestic students (Australian citizens) or humanitarian visa holders take out a loan from the government, which is paid back post-study upon earning a particular income threshold, deducted from one's income at each pay cycle [4]. Students who do not fall into these categories must pay semester fees by a census date for each semester, usually 4 to 5 weeks after each semester commences. As such, finances do not typically hinder domestic or humanitarian visa holder students from attending university.

## 2. Literature Review

### 2.1. School Experiences and Widening Participation

Due to recent changes in government aspirations to have 40% of 25–34 year olds with undergraduate degrees by 2020 [5], p. 6, there has been a shift in perception to include students that come from 'non-traditional' backgrounds if these goals are to be realised. It has been recognized that if university participation rates continue as they are, Australia is "rapidly moving towards a major skills crisis" [6], p. 31, and despite the growth in the higher education sector generally, this has not been accompanied by "increases in social equity" [6], p. 33. Students from non-traditional backgrounds have typically been "disenfranchised from university attendance" [6], p. 11 as their schools tend not to "have a focus on the academic curriculum required for university attendance" [6], p. 11. This is tied together with the geographical remoteness that some students can experience when living in areas that are distant from universities [6], p. 12, which requires that universities respond in more diverse ways to accommodate for more diverse potential students. Providing these opportunities is a matter of social justice [6], so that students from the "least advantaged ... especially ... lower socio-economic backgrounds" [6], p. 30, are able to determine their own futures through university attendance [7]. There is also a need for such

participation to be more "representative of all Australians" [6], p. 31, so that it incorporates people from all backgrounds to encourage more holistic economic growth to meet the aforementioned government goals [6], p. 37. This requires "creating spaces for them, not simply creating more places" [6], p. 43.

### 2.2. Non-Traditional Students' Engagement with Higher Education

One of the key issues in increasing university admissions for students from non-traditional backgrounds is their engagement with university while in high school [8]. School can be a risky environment for young people, particularly where they are not engaged in the educational process in secondary school, which can often lead to "economic risk, with a danger of lifetime exclusion from reasonable paid work or any paid employment at all" [9], p. 261. Students who come from low socio-economic backgrounds are more likely to attend lower performing schools [8], with parents and guardians that are not as familiar with the higher education system [8], p. 50. This lack of understanding of how the university system works, particularly how transition from high school to university takes place and the kinds of social and academic support that those under their care may need, does little to aid students' ability to consider higher education as a viable option for their future. Parents may not be aware of these realities due to their non-attendance at university themselves, especially if they come from lower socio-economic backgrounds and/or have less exposure to attending or learning about how universities function. As this is an unfamiliar environment for many students, the prospect of "fitting in" [10] in [8] p. 56 and even "hid[ing] potentially stigmatising attributes" [10] in 8 p. 56 that derive from their social class become real inhibitors to experiencing university life. Students from these backgrounds often understand themselves to be those that are "perceived as members of a devalued group" [8], p. 58. If these students are perceived as devalued, they are more likely to "withdraw from these [educational] settings" [8], p. 59. There is an emphasis in many universities to respond to the need to increase student participation, albeit without realising the lack of some students' "cultural and social capital to fulfil their aspirations" [11], p. 6 in [12] p. 122. Thus, widening participation should involve attracting students from non-traditional backgrounds, and, importantly, facilitate their overall success in higher education [13], p. 959. Demystifying academic culture, which can be a barrier to aspiring towards higher educational attainment due to a lack of understanding of how it works, is an important step to redistributing the power imbalances and providing non-traditional students with the cultural and social capital [13]. That is to say, it is important to make "the implicit explicit to students" [13], p. 949.

### 2.3. Student Disengagement and Higher Education

Student disengagement should not be treated as simply apathy from restless teenagers not wanting to engage with the learning process. Disengaging in the school process can lead to a compromise of students' ability to partake in a "more just distribution of resources and goods" [14] in 9, p. 261, which "is linked to notions of social justice" [14] in [9] p. 261. Consequently, the happiness of students who take part in the education process is compromised, which further affects students' desire and ability to aspire towards higher education. Some young people develop a disbelief in their ability to aspire towards meaningful employment in the future, which therefore places them at risk of not becoming active and contributing citizens to their societies [14] in [9] p. 261. Students' responses to different life stressors alongside their internal resilience often affects how they perceive their ability not only to complete high school, but to go onto higher education. Those "without school connections . . . have extremely limited opportunities to engage . . . in higher education" [15], p. 23, despite the ongoing aspirations of students that come from low socio-economic backgrounds [16]. These processes of disengagement can be challenged by positive engagement with "a trusted adult" [17] in [18] p. 119, and where engagement between students and teachers is ongoing and intentional; this can radically shift a young person's involvement in their schooling [18].

Factors that assist school students aged 14 to 19 years to engage meaningfully with the learning process have also been explored by scholars [18]. It was found that disengagement in school occurs

as a result of three overriding factors: the depth of relationships shared peer to peer and teacher to peer; the quality of the teaching; and the perceived relevance of more "practical/vocational subjects compared with academic subjects" [18], p. 113. These factors lead to "[a] defined relatedness with respect to school climate, teacher relationships, feelings of belonging and acceptance and inter-personal support" [18], p. 113, all of which contribute to how students engage with their studies.

### 2.4. Comparative Evaluations of Widening Participation Programs

Some of the evaluations that have taken place in Australia include the National Centre for Student Equity in Higher Education or NCSEHE's evaluation of 31 widening participation initiatives around the country [19]. These programs and support services are funded in Australia by the Higher Education Participation and Partnerships Program (HEPPP) funding scheme. The NCSEHE reported on the importance of university partnerships, and how successful outcomes were ascertained throughout these projects. Some of the goals of these projects include offering support to students from non-tradition backgrounds, interventions to assist accessing university courses via bridging and other programs, and encouraging them to maintain and complete their studies [19].

Widening participation programs at one Australian university target non-traditional students, and have developed perspectives towards more inclusive practices within widening participation [20]. Other scholars identified key components that increase the effectiveness of widening participation programs, namely, meaningful collaboration between communities and stakeholders and university staff [21], p. 40; on-campus experiences, such as information sessions and campus tours [21], p. 41; mentoring from university students and academics, who act as role models [21], p. 41; and collaborative learning at school, where widening participation staff engage with schools to develop and deliver relevant and aspiration-building content taught in schools [21], p. 43. In terms of evaluation, students' confidence to come to university, study a particular course of study (e.g., science), and increased knowledge around alternative means of university entry were increased, amongst others [21], pp. 37–39.

Similarly in the U.K. and U.S., widening participation programs target students from non-traditional backgrounds [22]. Evaluations here are conducted by using the "counterfactual condition"—realising the impact and changes produced by the program by comparing it to similar settings without a program [22], p. 744. Randomization and quasi-experimental designs are also given as effective evaluative models [22].

Widening participation is the "key mechanism" where universities can reach non-traditional students and promote university attendance, however, evaluation is generally scarce and "little is known about their effectiveness" [23], p. 1384. There is a need for a greater commitment to evaluating widening participation programs [24]. Evaluation of widening participation programs is critically important to "facilitate both programme development and the creation of an evidence base guide to future practice and policy" [24], p. 384. This article addresses this imperative in the literature by considering how the FFP functions to increase aspirations for the students it works with, and how the program fosters these aspirations alongside relationships between the different stakeholder groups, which collectively assist in highlighting university as a viable option for the students that engage with the program.

### 3. Methodology

Social Ventures Australia [25] developed the SROI methodology, which is currently being used by a range of organizations to assess the social impact of their programs. This methodology was chosen as a relevant and meaningful tool as it measures impact at multiple levels when compared to typical cost-benefit analyses, and features a strong narrative element in the incorporation of the assessment of social impact and change. Stakeholders and not researchers determine the elements of the program that are considered most valuable, and the programs are therefore evaluated for the social purposes they are serving [26]. Both qualitative and quantitative measures contribute towards the development of a cost to benefit ratio that monetizes immaterial social changes that took part due to the FFP taking place, such as increased aspiration to attend university and increased confidence on campus. SRIO's ability to monetize these attributes is unique, and provides a more holistic account of how change

happened within the communities that engaged with the FFP. The strong narrative focus also allows for a fuller understanding of how the methodology captures change, and was therefore considered the most relevant methodological tool for the current research project.

The SROI methodology has been described as

- A means of measuring the social impacts of projects, programs, organizations, businesses, and policies;
- A form of stakeholder-driven evaluation blended with cost-benefit analysis tailored to social purposes.

Part of the goal of this methodology is to create room for initiatives that seek to reduce poverty for all involved, be it economic, social, opportunistic, or other. This process seeks to place stakeholders (i.e., participants) at the centre of the evaluative process [27]. This methodology is considered an important means of assessing the performance of philanthropic social projects, especially where solely quantitative measures fall short when addressing improvements in concepts such as increased self-confidence, self-determination, and motivation and aspirations towards higher education, which is a critical aspect of the FFP.

This methodology has three key performance indicators: appropriateness, effectiveness, and efficiency [26], and is undergirded by seven principles:

1. **Involve stakeholders**. Stakeholders (i.e., participants) inform what is considered meaningful to the FFP within their context and how the program is measured and valued.
2. **Understand what changes.** Describing how change has happened as a result of the FFP. This change is not limited to positive and intended change, but also includes negative and unintended change. These changes are understood as outcomes, which form a theory of change—how FFP has impacted the community, and how these changes are measured for all participants. Each outcome is measured through indicators, validating their effectiveness.
3. **Value the things that matter.** Outcomes such as increased aspiration and motivation are given financial proxies in order to value how the outcomes can be measured quantitatively.
4. **Only include what is material.** Deciding what information needs to be included when assessing the program, so that an accurate picture of the program's impact can be ascertained. This step uses the term 'materiality', where "information is material if missing it out of the SROI would misrepresent the [program's] activities" [26], p. 9.
5. **Do not overclaim.** This analysis claims only what FFP is responsible for. An example of overclaiming would be attributing an increase in motivation to the program when in fact students' parents have continuously encouraged university participation throughout their schooling careers. Deadweight refers to the amount, expressed as a percentage, of the outcomes that would have happened without Fast Forward being present in the school. Attribution is similar to deadweight, except that it measures if/how these outcomes would have been met by other people (students, parents, teachers, other influences). The final consideration in this section is drop-off, which measures how the impact of the outcomes depreciate over time, especially the outcomes that are derived solely from the FFP. It may be that, over time, other factors or influences impact the participants more than the program itself. Drop-off is considered for outcomes that last for more than one year; this analysis considers the impact of the program for only one year (2017), and thus drop-off was not applied in this analysis.
6. **Be transparent.** Reflect the honest perspectives of research participants, as well as the authors' position as those who want to show FFP to be an effective program. At this point, theory of change is developed, showing the processes of change that have occurred, and how this has led to outcomes being met. The SROI ratio of money invested—money gained, on the basis of the commodification of the outcomes as material—is also developed at this point, which highlights the economic impact of the program whilst drawing upon the social, educational, and other outcomes that have been delivered as a result of the program.

7. **Verify the result.** Present findings to all stakeholders via emailing a two-page report to school representatives, which was distributed to those that took part in the research.

Each step of the methodology will now be expounded to show how the principles of SROI highlight the effectiveness of the program. Following the format of the survey, step 2 (understand what changes) was switched with step 3 (value the things that matter) to make for a more chronological reflection of the findings.

Human ethics approval was sought and granted through Western Sydney University, and additionally through the New South Wales Department of Education, WSU ethics number H12079.

### 3.1. Step 1: Involve Stakeholders

The FFP has been running since 2004 in 63 high schools across greater western Sydney, and, in 2018, expanded to 81 schools. Considering the vast number of schools the program partners with, it was decided by the research team to select six schools that had been in the program for at least five years, to see how the longevity of the program has affected participants' involvement and perceptions of the program. The schools selected to take part in the programs had a mixture of cultures, with some schools having most of their student body coming from migrant backgrounds. Table 1 shows how much time each stakeholder (that is, participant) group spends in the program. The term 'year 9' or other number means the year of education attained. In Australia, high school commences at year 7 (11–13 years of age) and finishes at year 12 (16–18 years of age). A total of five of the six nominated schools responded and took part in the survey, with 42 year 9 responses, 26 from year 10, 19 year 11 responses, and 13 from year 12, alongside 4 parents, 3 project officers, and 2 university staff members, totalling 109 survey responses.

**Table 1.** Levels of involvement for stakeholders.

| Year | Activities in Program |
|---|---|
| Years 9–11 | One full day on campus (9:30–2:30); two to four workshops in school (1.5 h); Year 9 'Welcome to Western Evening' (2 h). |
| Year 12 | One full day on campus (9:30–2:30); two to four workshops in school (1.5 h). Optional access to Higher School Certificate (HSC) preparation/attendance at Western Sydney University and open days. |
| Parents | Totalling 1 'Tertiary Information and Pathways' session; 8 'Welcome to Western' evenings (Year 9 parents); 12 evenings for parents of year 10–12 students (four workshops respectively for parents of each year). |
| Project officers | Four full-time (35 h per week) professional staff, comprising portfolios capturing an even division of the 63 of schools. |
| WSU support staff | Year 12 conference sessions. |

Participants were asked to complete a paper (for students) and online (for all other groups) survey that asked questions around their personal background (year group, gender, school attended/length of time in currently employed position, relationship to university (did you/your parents attend university, do you/your parents encourage you to go to university), as well as their perceptions of the best parts of the program (qualitative: what are the best parts of the program, what have you learned from being involved?). The remaining three questions asked about the value of completing school and balancing time, alongside responses to the changes and improvements that occurred as a result of the program, to be discussed below (see value the things that matter, pp. 14–24 below).

The research team gave the selected schools consent forms and information sheets, which the schools in turn gave the year 9–12 students to take home to obtain parental consent to take part in the evaluation. Those students who returned a signed consent form were able to complete the paper survey. One of the research team members negotiated a time and date to visit the school, with teachers

from the schools organising for students to be present in particular classrooms so that they could complete the evaluation. The research team member met the students, provided a verbal explanation of the research and what they were being asked to do, and completed the survey. All other stakeholder groups were emailed an invitation to take part in the research, and this email contained a website link to the survey to complete. Schools contacted parents whose children took part in the FFP with this same email with the survey to invite them to take part.

Some respondents completed the first question (consent to take part in the research) without completing the remaining questions. This is why, despite the stated numbers of participants here, the total number of responses explored in this article do not always align with the number of participants in each group. Tables 2 and 3 show the demographic and educational information pertaining to each stakeholder group. As some questions were only asked to specific stakeholder groups, a few of the entries appear blank.

**Table 2.** Demographic and educational information for student stakeholder group.

| Stakeholder Group | How Many? | Gender | Did Your Parent Attend University? | Do Your Parents Encourage You to Attend University? |
|---|---|---|---|---|
| Year 9 | 42 | 19 (47.50% male), 21 (52.50%) female | Mothers: 5 (12.50%) Fathers: 6 (15%) Both parents: 5 (12.50%) Neither: 11 (27.50%) Unsure: 13 (32.50%) | A lot: 24 (60%) Somewhat: 14 (35%) Not really: 2 (5%) |
| Year 10 | 26 | 11 (44%) male, 13 (52%) female, 1 (4%) other. | Mothers: 0 (0%) Fathers: 4 (15.38%) Both parents: 2 (7.69%) Neither: 17 (65.38%) Unsure: 3 (11.54%) | A lot: 12 (46.15%) Somewhat: 8 (30.77%) Not really: 6 (23.08%) |
| Year 11 | 19 | 5 (26.32% male, 14 (73.68%) female. | Mothers: 3 (16.67%) Fathers: 3 (16.67%) Both: 2 (11.11%) Neither: 10 (55.56%) | A lot:13 (68.42%) Somewhat: 6 (31.58%) |
| Year 12 | 13 | 3 (23.08%) male, 10 (76.92%) female. | Mothers: 3 (25%) Fathers: 0 (%) Both: 2 (16.67%) Neither: 7 (58.33%) | A lot: 9 (75%) Somewhat: 3 (25%) |

**Table 3.** Demographic and educational information of parents and staff stakeholders.

| Stakeholder Group | How Many? | Gender | How Long Have You Held Your Current Position? | Did You Encourage Your Child to Attend University Before the Fast Forward Program (FFP)? |
|---|---|---|---|---|
| Parents | 4 | 4 (100%) female | | A lot: 1 (33%) Somewhat: 1 (33%) Not Really: 1 (33%) |
| Project officers | 3 | 1 (33.33%) male, 2 (66.67%) female | 0–2 years: 3 (100%) | |
| WSU support staff | 2 | 2 female (100%) | 0–2 years: 1 (50%) 3–4 years: 1 (50%) | |

### 3.1.1. Year 9

Parents in this group wanted their children to attend university, with 95% of parents encouraging their children 'a lot' or 'somewhat'. However, 32.50% of students were unsure if their parents attended university; parents seemed not to be telling their children about their achievements. When combined

with the 'no' response, 24 of the 42 respondents, or 60%, had parents that either did not engage with university or do not talk about their experiences with their children.

Therefore, from these results it is deduced that parents desire better outcomes for their children, especially from the schools with a high population from migrant backgrounds. It is known that parents are the strongest influence upon children matriculating into higher education [28], and the participants' responses here affirm this point.

### 3.1.2. Year 10

Just as with Year 9, there were clear aspirations for wanting to attend university; 76.92% of parents encouraged their children 'a lot' and 'somewhat'. Parents had not, however, been as exposed to the university system in this cohort, emphasising the need that the FFP is addressing between aspiration, exposure, and the realisation of students' potential to make university study a part of their viable future options.

### 3.1.3. Year 11

Again, as with the former year groups, there was a clear desire to access higher education, without necessarily having the tools to do so. No parents discouraged university attendance, and 65.38% of participants had neither parents attend university. The FFP bridges a gap that is sorely needed for these students to successfully matriculate into higher education, especially university.

### 3.1.4. Year 12

There was a strong correlation between the lack of educational attainment from parents and the desire to have their children take part in it. Year 12 further highlighted this correlation, with 55.56% of participants having neither of their parents attend university and no parents in the study discouraging attendance. It is known that parents are the single greatest influencers upon a child to attend university [29], but where parental knowledge of university systems is not strong, university staff as "trusted adults" [17] in [18] p. 119 can help to bridge this gap more effectively.

### 3.1.5. Parents

It is important to recognize that only female parents responded to the survey. Mothers are influential upon their child's decision to go on to further education and training [30], and their participation in this survey reflects this reality. The responses here showed an even distribution across the response range, which is different from the responses that students gave concerning their parents.

### 3.1.6. Project Officers and Western Sydney University Support Staff

This group was composed of paid staff who are actively involved in facilitating workshops and other key components of the FFP. As part of the SROI process, it is important to also capture workers' perspectives on the value associated with their contribution in the program, and how this impacts on achieving key outputs and outcomes.

## 4. Findings

### 4.1. What Are the Best Parts of Being Involved in the FFP?

### 4.1.1. Students

Reponses to this question were collated and reflected in the theory of change map (see pp. 22–28; Figures 1–7 below). Some of the participants' individual voices emphasized the program's impact. As three year 9 students noted,

> *Fast Forward is a program that looks out for students like myself who is [sic] worried about their university/future.*

> *I believe that I can achieve university and that it is not a scary place.*

> *Giving me the opportunity to succeed and reach my full potential and to show that disability doesn't prove anything and to take on anything that comes your way.*

Students stated that the program gave them the experience of being on campus and an opportunity to engage with a university program, which was a new experience for most. They were also able to access help with study skills and resources, and grew in confidence to go to university. Some students realized that university was a viable option for them. The program is therefore addressing a clear need for these students, giving them a more confident expectation of and relationship with higher education, and developing study and resource-accessing skills that will help their futures, even if they do not decide to study at university.

The research also indicated an increase in understanding and confidence to go on to higher education. As two year 10 students stated,

> *It motivates me to get great marks to get into university, and it also builds my leadership, communication skills and confidence.*

> *Learning alot [sic] of different skills and learning alot [sic] about the University. Before I attended this program, I was kind of scared about university but now I'm very confident thanks to Fast Forward.*

Confidence has increased through the FFP, and students have expanded their perspectives of what their futures might hold.

A common reflection amongst the students was the newfound awareness of what university life is like and how going onto campus was valuable to them. They also enjoyed asking questions and learning about opportunities that university presents, and they gained an increased knowledge about different career paths:

> *We are experiencing how university life is like and the teaching. There is always someone to guide you and you can ask them questions about university.*

> *We get to learn about university, I did not know what I wanted to do but now I have somewhat of an idea. We learn and we have fun. Fast Forward helps me with my future.*

The FFP is equipping students with an ability to develop foresight for their futures, and allows them to consider what university might be like.

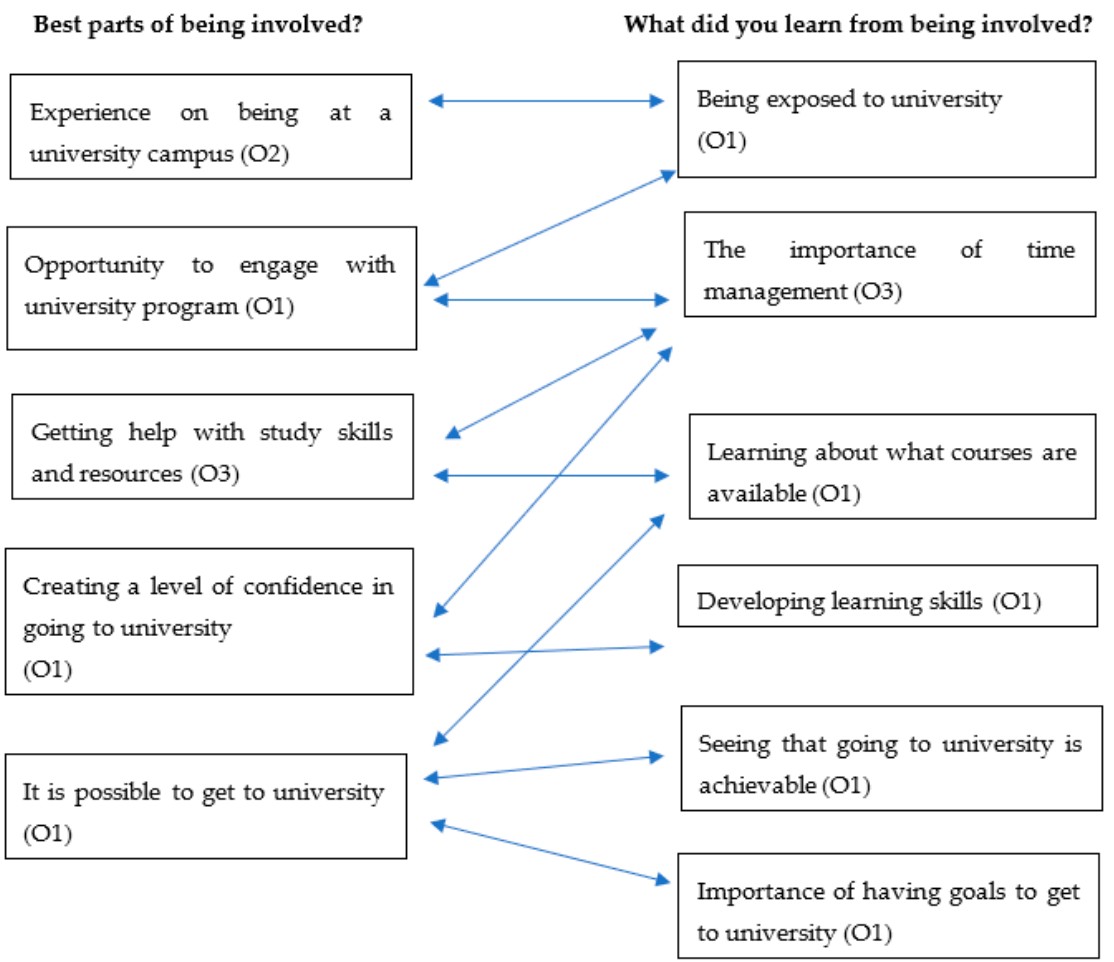

**Figure 1.** Year 9 theory of change. O: outcome.

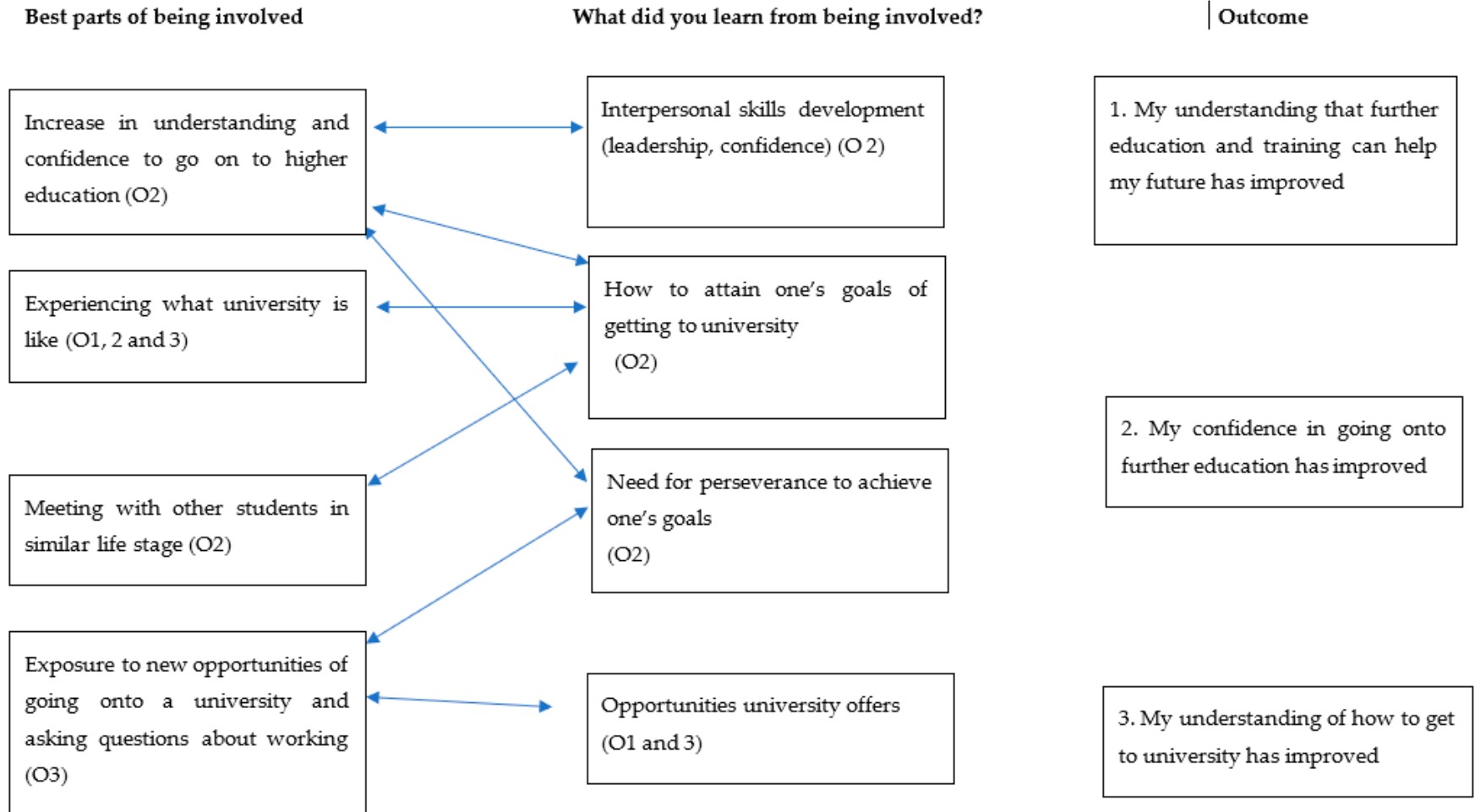

**Figure 2.** Year 10 theory of change.

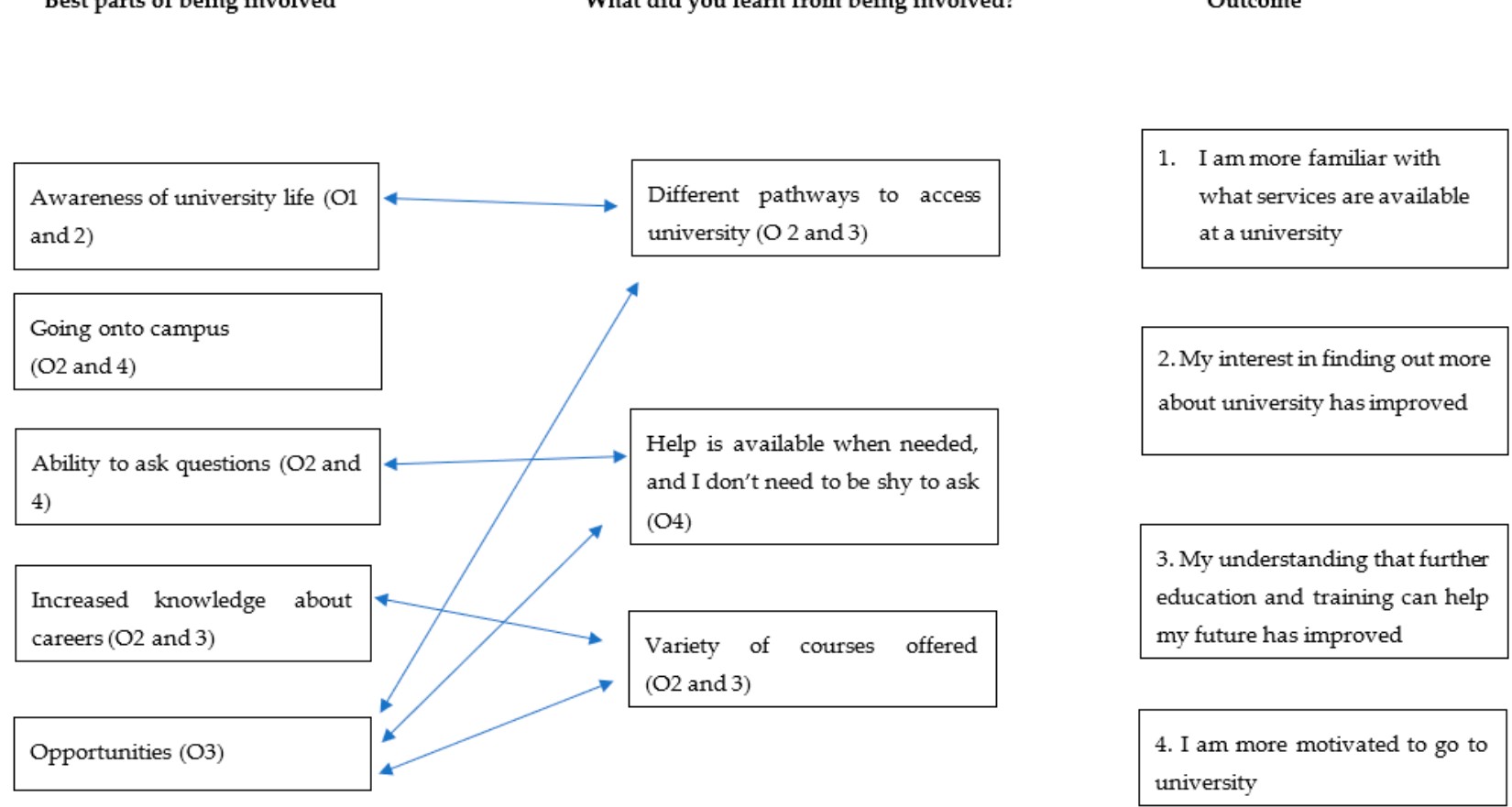

**Figure 3.** Year 11 theory of change.

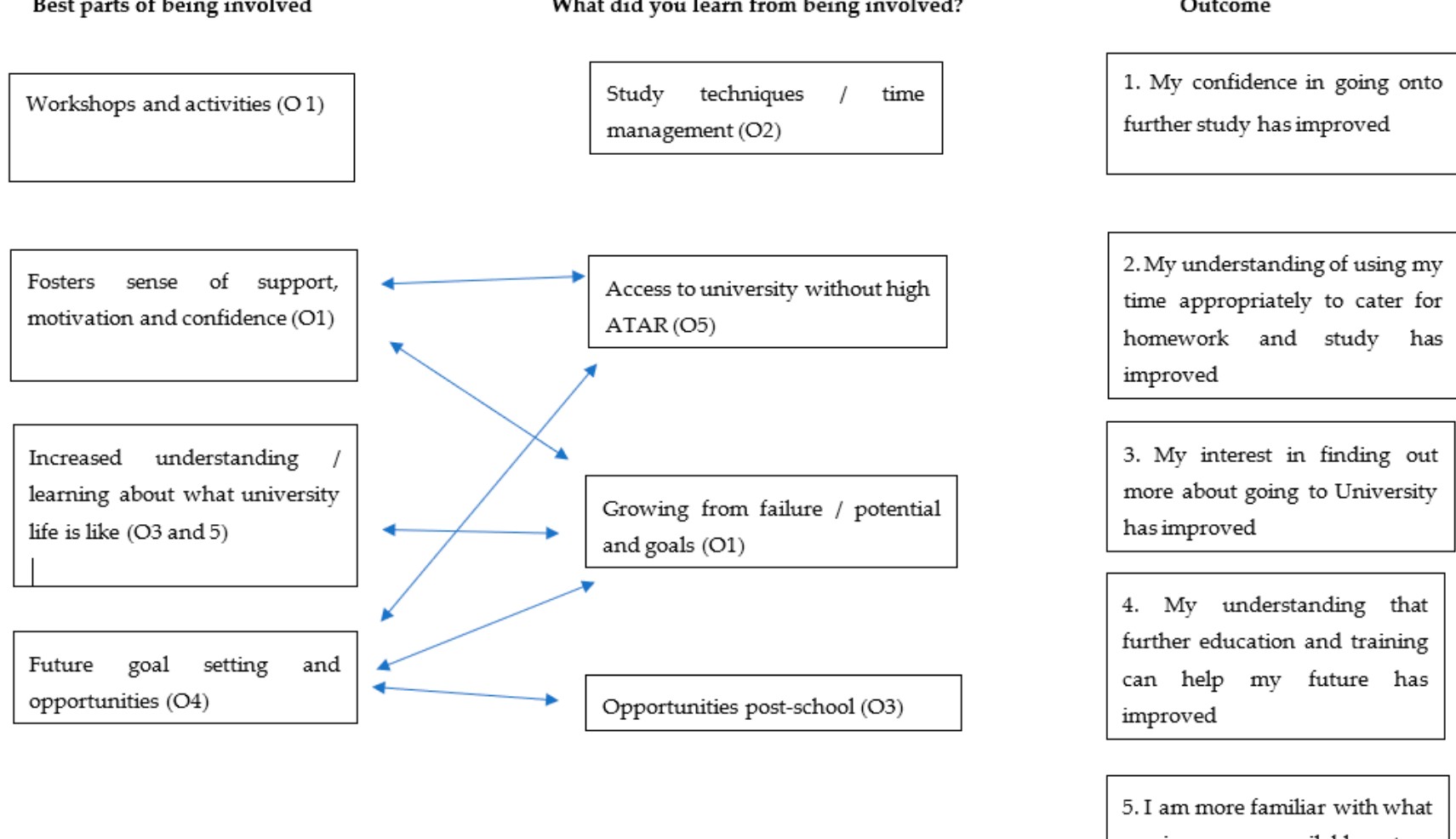

**Figure 4.** Year 12 theory of change.

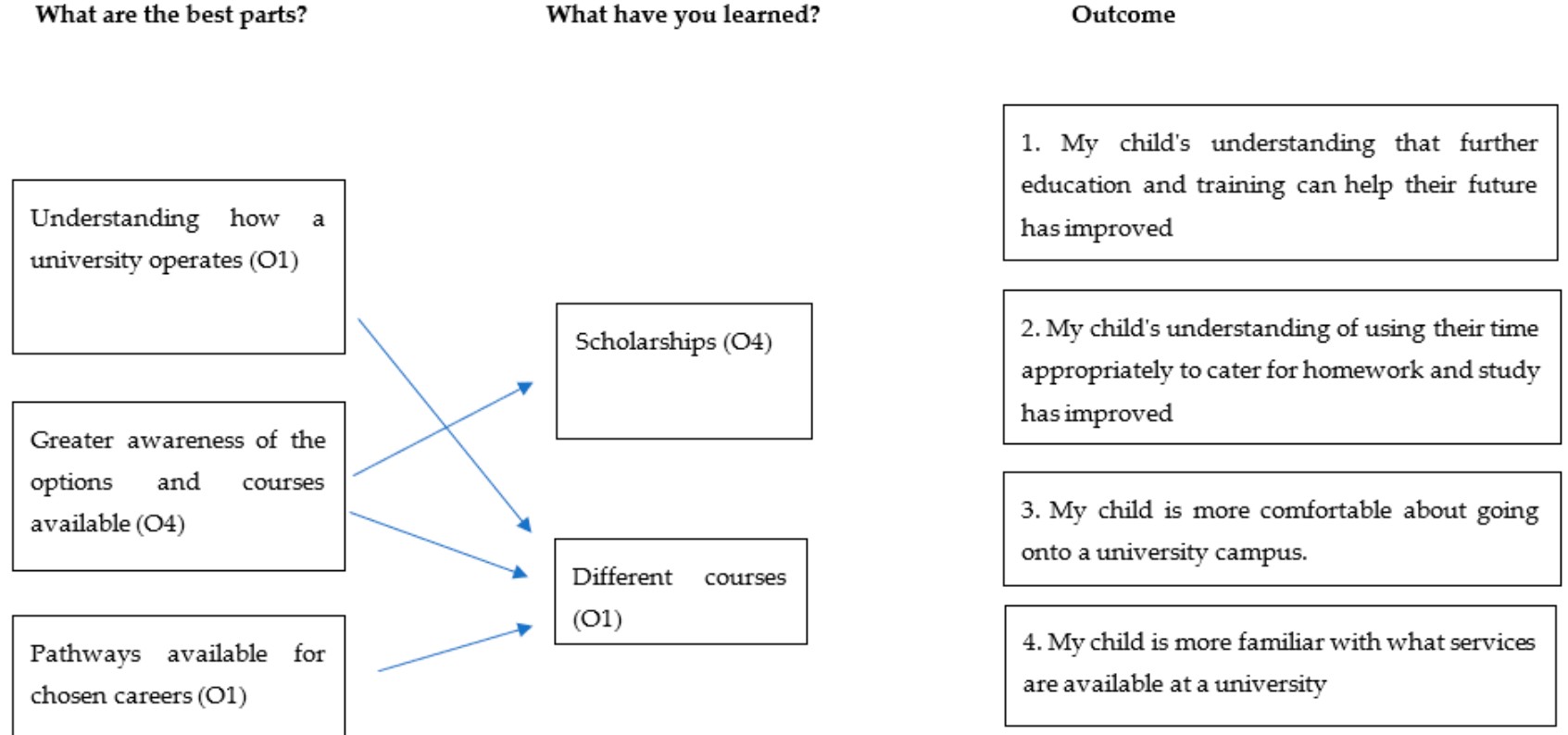

**Figure 5.** Parents' theory of change.

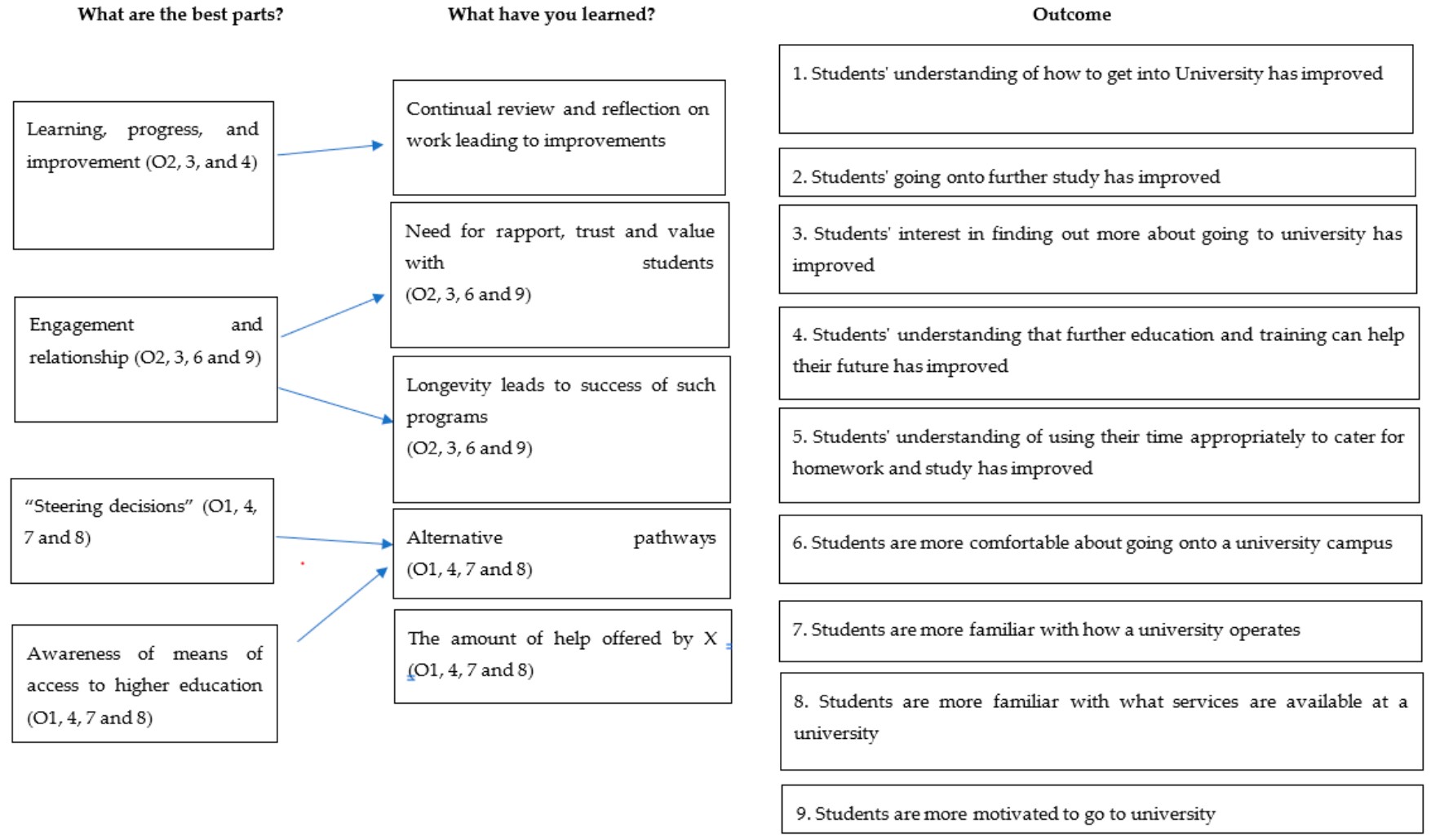

**Figure 6.** Project officer theory of change.

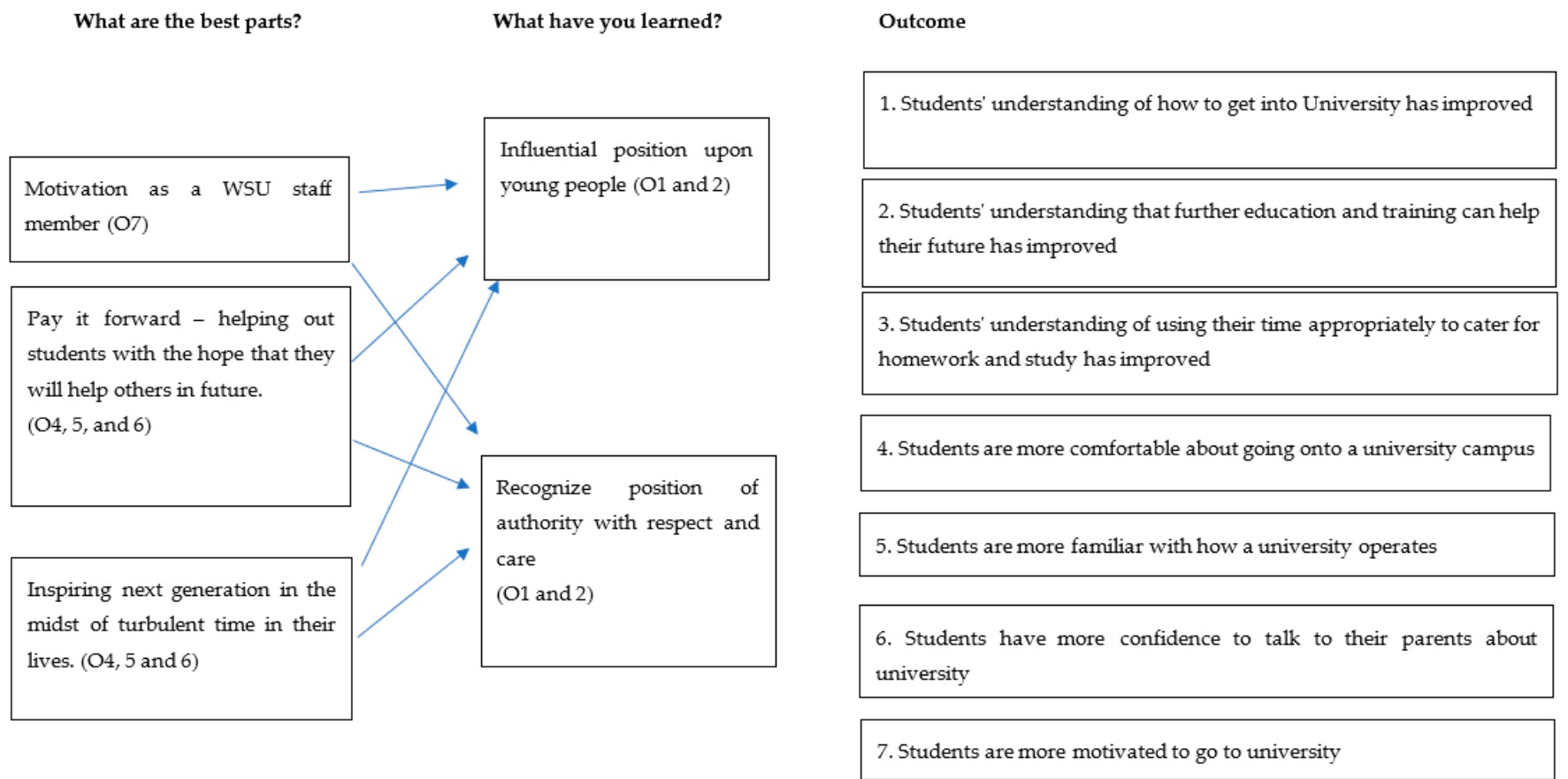

**Figure 7.** WSU support staff theory of change.

Students reported that the workshops and activities were beneficial, which indicates that they gain a sense of support, motivation, and confidence through their participation in the FFP. The program also encouraged students to set goals for their futures. This was particularly noteworthy for year 12 students,

*Fast Forward provides a massive support system in my opinion. It's incredibly encouraging and motivates me to graduate and make it to uni. I also really enjoy the workshops, they provide knowledge you can't just google to learn about.*

*Being able to gain an insight into a true Uni lifestyle. Having the opportunity to visit the campus and experience lectures and talks. Gaining a sense of motivation for not only Uni life, but other aspects of general life.*

The acquisition of knowledge that pertains to accessing university and other 'aspects of general life' were celebrated by these students, showing that the program is valuable to their academic and personal aspirations.

### 4.1.2. Parents

Parents valued the program giving students a greater awareness of the options and courses that are available to their children, and the different pathways they can take for different careers. In their own words,

*Getting to see how the uni is set-up [sic] and finding out about courses available for my daughter.*

*My daughter has a better understanding of University life and how to navigate her pathway to her chosen career.*

The FFP is exposing students to opportunities and knowledge that they otherwise would not have received from their parents alone, providing a kind of surrogate connection between adults that can help raise these aspirations for parents that have little or no awareness of the options that university can present to their child.

### 4.1.3. Project Officers

Project officers highlighted the importance of engagement and relationship when working together with school students; the need to learn, progress, and improve the program and their delivery of it; the power of "steering decisions" for the young people they assist; and awareness of students' needs and their own professional practice. Project officers stated,

*Seeing students really connect with a particular activity or actively engage in the learning process is most rewarding. Witnessing staff appreciation for the work we do reinforces the value of the programs.*

*The best part of the . . . program is developing a relationship with participants over the four years they participate in the program and noticing their progress and improvement . . . I feel that I can have a significant impact on . . . [their] career[s] and steering decisions . . .*

*Watching the students' faces when they realise it is not that difficult and that others have overcome difficulty to get to uni. Hearing comments like, "now I know what I want to do" and "this is the first time I have looked up different jobs and thought about them".*

The FFP therefore provides a process of identification for these young people, where they are able to develop working relationships with project officers that can assist them, become a trusted face over the course of the four-year program, and encourage their educational and personal goals in a way that is individualized and meaningful.

4.1.4. Western Sydney University Support Staff

Western Sydney University staff members saw their involvement as motivating, a means of 'paying forward' what they had gained from their education, and an opportunity to inspire the next generation during a potentially turbulent time of life. Support staff said,

*You are reminded why you began your own journey, you have a chance to pay it forward and tell your younger self a few things you wish someone had told you. You are involved in possibly inspiring someone great during their more confused and younger years.*

*Not only does the program build aspirations for young people to access and be successful at their current schooling and university, but it also creates a platform that celebrates and validates the experiences of academic staff to share their life journeys with the next generation.*

*4.2. What Did You Learn from Being Involved in the FFP?*

4.2.1. Students

Exposure to new forms of knowledge was repeatedly mentioned by students:

*I have learnt alot [sic] from Fast Forward, especially from the online website (YourTutor, now known as Studiosity, is an online tutoring service that connects students to teachers to help with Mathematics, English, Science and other subject areas. More information can be found at https://www.studiosity.com/.) helped me with my exams and NAPLAN [National Assessment Program – Literacy and Numeracy].*

*The Fast Forward Program has taught me how to balance school and life.*

*I have learned that there are many opportunities for us when we go to university. I have also learned that the uni staff care about young students studying and education.*

*To work with others to be independent[,] develop my life skills[,] have tudors [sic] to help me whenever I need them.*

Students were more aware of university support systems as a result of engaging with the FFP. The use of the online homework and tutoring service 'Studiosity' can be accessed whilst still in school, which promotes a greater likelihood that students internalize a sense of self-directed learning, leading to an attitude of lifelong learning that will hopefully persist for many years.

Moreover, the program promoted students development of interpersonal skills such as leadership and confidence, the ability to attain one's goals, and the need for perseverance to achieve them, as well as the opportunities that university offers to be of great value:

*I learnt so many spectacular things about university, I learnt how to be more involved as well as making new friends and I also learnt that going to university is the one of the greatest achievement especially in our western Sydney region.*

*I have learned that going to Uni is hard and I have to study hard and get good mark and if I try my best it's not gonna be hard and impossible as I though[t] it would be.*

*The Fast Forward Program has taught me how to interact and participate more, it also taught me how to achieve my goals in order to achieve my dream job.*

Through repeated and consistent participation in the FFP throughout years 9–12, the need for skills development that pertains to high school success and eventual matriculation to university or higher education more generally are emphasized. The development of these skills fosters an internal sense of ability and capability for their futures.

The importance of different pathways to access university was appreciated by students in years 11 and 12. The provision of help encouraged students to be more proactive and hesitate less when they needed assistance. They were also helped by the variety of courses available at university.

*I've learned how to be capable with everyday life . . . Doing harder in school work and achieve my goals or getting high ATAR (Australian Tertiary Admission Rank, which is the primary factor used to determine university entry in Australia, showing a comparative rank between all Year 12 students. It is expressed as a two-digit, two decimal figure, with 99.95 being the highest score).*

*To be a good leader, to understand all aspects of how to stay focussed. How to further myself.*

Year 11 and 12 students also learnt about accessing university without a high ATAR; growing from failure; and how such growth can lead to the development of potential and goals, study techniques, and time management:

*There is more than one way to achieve your goals and failing is okay as it helps us grow and make us appreciate things a lot more.*

*I've learned that university isn't as difficult as I've been told. I've also learned to recognise and use my great abilities. I've also learned that failure isn't bad, in fact it's NECESSARY in order to succeed.*

*Different opportunities available for life after school which has heavily impacted my view and understanding on what I am capable of.*

Perhaps at the time when they need it most, these students have shown how the FFP encourages them to gain insights into what they are able to do with their futures, and not take mistakes or failure as personally as they may have prior to attending these workshops. As the students pointed out, the program mitigated the stress and pressure of year 12. Resilience and fortitude are promoted through the program, and these are essential characteristics no matter which path is selected for life after school.

### 4.2.2. Parents

Parents learned that scholarship opportunities and different courses were available for their children. One parent stated they had learnt "not much to be honest", despite there being parent information nights available through the FFP. Again, parents are learning about what universities have on offer, and are therefore better equipped to be able to encourage their children to make the most of these opportunities.

### 4.2.3. Project Officers

Various concepts were shared here, namely the importance and value of continual review and reflection upon one's work, leading to improvements; the need to establish rapport, trust, and value with students; the fact that longevity in such programs leads to them being successful; the importance of sharing information pertaining to alternate pathways to access university; and the amount of help offered by Western Sydney University. As an institution, Western Sydney University seeks to engage with young people who have aspirations, but often not the social and educational capital to access university firsthand, therefore making the process of course selection easier when applying to university. As one Project officer stated,

*I have been reminded of how little careers ed[ucation] is done before Year 12 in some schools, which means many students are very confused at subject selection time. Their ability to choose subjects to suit their interests competes with pressure to pick subjects for a good ATAR or for uni, when they may change which course they want to do. Schools vary greatly in the quality of the Careers Advisor.*

Where schools do not address directly the concerns of which subjects to select and how each student might engage with a career of their choosing (which has been seen to be of prime importance across all stakeholder groups in this analysis, see Section 4.3 below), the FFP helps to assist students at this point of time and in a timely fashion.

#### 4.2.4. Western Sydney University Support Staff

One of the Western Sydney University staff members responded by saying,

*We hold a position of trust and authority and to treat that with respect and care when influencing young minds.*

The participation of such Western Sydney University staff members therefore creates a reflective/reflexive space for current staff, fostering a sense of purpose and recognition of their position, which this staff member sought to use in a benevolent and altruistic manner.

### 4.3. Step 2: Value the Things That Matter

Valuing what matters was ascertained through quantitative responses to the questions listed in the Table 4 matrix response selection (not really, somewhat, a lot, not sure) to the question "How important are the following to you?". The top three answers for each stakeholder group are stated below. Various sections of the table remain empty as the data within highlights only the top three answers for each stakeholder group.

**Table 4.** How important are the following to you?

| Question | Response | | | |
|---|---|---|---|---|
| | **A Lot** | **Somewhat** | **Not Really** | **Not Sure** |
| Completing high school by finishing year 12 | Year 9: 35 (89.74%)<br>Year 10: 22 (84.62%)<br>Year 11: 19 (100%)<br>Year 12: 12 (100%)<br>Project officers: 3 (100%) | Year 9: 4 (10.26%)<br>Year 10: 4 (15.38%) | | |
| Finishing school before year 12 to get a job | | | | |
| Going to university | Year 9: 28 (71.79%)<br>Project officers: 3 (100%)<br>WSU support staff: 1 (100%) | Year 9: 9 (23.08%) | Year 9: 2 (5.13%) | |
| Going to TAFE/college | | | | |
| Getting into a job you are passionate about | Year 9: 35 (89.74%)<br>Year 10: 23 (88.46%)<br>Year 11: 17 (89.47%)<br>Year 12: 12 (100%)<br>Project officers: 3 (100%)<br>Parents: 3 (100%) | Year 9: (10.26%)<br>Year 10: 3 (11.54%)<br>Year 11: 2 (10.53%) | | |
| Supporting your family | Year 9: 31 (79.49%)<br>Year 11: 18 (94.74%) | Year 9: 8 (20.51%)<br>Year 11: 1 (5.26%) | | |
| Participating in other sport and community commitments (e.g., church/cultural groups/dance, etc.) | Project officers: 3 (100%) | | | |
| Being able to balance study, family, and community commitments (including household chores) | Year 12: 12 (100%)<br>Parents: 3 (100%)<br>Project officers: 3 (100%)<br>WSU support staff: 1 (100%) | | | |
| Being able to balance study and paid work commitments | Year 10: 18 (69.23%)<br>Parents: 3 (100%)<br>Project officers: 3 (100%)<br>WSU support staff: 1 (100%) | Year 10: (30.77%) | | |

### 4.3.1. Year 9

The year 9 cohort have less pressing immediate needs than their counterparts, such as the need for balancing time, and were able to absorb more of the long-term aspirational goals of the program (finishing year 12 and going to university).

### 4.3.2. Year 10

These priorities are not surprising, given the increasing focus on the need to work and complete school in order to have as broad an opportunity base as possible going into the future.

### 4.3.3. Year 11

These year 11 students were surveyed in term 4, 2017, and these outcomes reflect where their focus is—completing their secondary schooling successfully and transitioning to a meaningful job that can help support their families.

### 4.3.4. Year 12

These responses are to be expected from students that were, at the time of being interviewed, on the verge of doing their final secondary school exams.

### 4.3.5. Parents

There was a clear focus on the need for successful and enjoyable future careers, and a focus on time management and priorities that are generally consistent with the aforementioned student priorities.

### 4.3.6. Project Officers

Project officers considered educational aspirations and the co-requisite need of strong time management skills as most important for their students.

### 4.3.7. WSU Support Staff

Again, there was a strong focus on accessing university and time management skills, a consistent theme across all stakeholder groups when considering what is important to them.

### 4.4. As a Result of Being in the FFP, What Areas Have Improved and What Areas Have Changed?

These two questions were used to determine the outcomes for each group by choosing the top three responses for each stakeholder group. The answers to the 'a lot' and 'somewhat' responses were combined when determining the top three answers, which were seen to indicate where the most improvement/change had occurred, and therefore highlighted the most significant impacts of the program for the relevant stakeholder group. Some questions had a higher 'a lot' response than the combined 'a lot' and 'somewhat' score; again, the aim was to capture how the program affects the most overarching sense of improvement/change, rather than only focussing on the most potent improvements/changes that had taken place.

Where more than three responses had the same level of positive response, they were included as outcomes, as seen in the parent, project officer, and Western Sydney University (WSU) support staff responses.

These responses are recorded in Table 5.

**Table 5.** Areas that have improved and changed.

| Statement | Response | | | |
|---|---|---|---|---|
| | **The Following Areas Have Improved** | | | |
| | *A Lot* | *Somewhat* | *Not Really* | *Not Sure* |
| My understanding of how to get into university | Year 10: 19 (73.08%)<br>Project officers: 3 (100%)<br>WSU support staff: 1 (100)% | Year 10: 7 (26.92%) | | |
| My understanding of how to get into TAFE/college | | | | |
| My confidence in going onto further study | Year 10: 20 (76.92%)<br>Year 12: 12 (100%)<br>Project officers: 3 (100%) | Year 10: 6 (23.08%) | | |
| My interest in finding out more about going to university | Year 11: 14 (73.68%)<br>Year 12: 11 (91.67%)<br>Project officers: 3 (100%) | Year 11: 4 (21.05%)<br>Year 12: 1 (8.33%) | | **Year 11: 1 (5.26%)** |
| My understanding that further education and training can help my future | Year 10: 22 (84.62%)<br>Year 11: 14 (73.68%)<br>Year 12: 11 (91.67%)<br>Parents: 2 (66.67%)<br>Project officers: 3 (100%)<br>WSU support staff: 1 (100)% | Year 10: 4 (15.38%)<br>Year 11: 4 (21.05%)<br>Year 12: 1 (8.33%)<br>Parents: 1 (33.33%) | Year 11: 1 (5.26%) | |
| My understanding of using my time appropriately to cater for homework and study | Year 9: 25 (64.10%)<br>Year 12: 12 (100%)<br>Parents: 2 (66.67%)<br>Project officers: 3 (100%)<br>WSU support staff: 1 (100%) | Year 9: 11 (28.21%)<br>Parents: 1 (33.33%) | Year 9: 3 (7.69%) | |

**Table 5.** *Cont.*

| Statement | Response | | | |
|---|---|---|---|---|
| | **The Following Areas Have Changed** | | | |
| | *A Lot* | *Somewhat* | *Not Really* | *Not Sure* |
| I am more comfortable about going onto a university campus | Year 9: 25 (64.10%)<br>Parents: 2 (66.67%)<br>Project officers: 3 (100%)<br>WSU support staff: 1 (100)% | Year 9: 12 (30.77%)<br>Parents: 1 (33.33%) | Year 9 2 (5.13%) | |
| I am more familiar with how a university operates | Year 11: 10 (52.63%)<br>Project officers: 3 (100%)<br>WSU support staff: 1 (100)% | Year 11: 9 (47.37%) | | |
| I am more familiar with what services are available at a university | Year 12: 11 (91.67%)<br>Parents: 2 (66.67%)<br>Project officers: 3 (100%) | Year 12: 1 (8.33%)<br>Parents: 1 (33.33%) | | |
| I have more confidence to talk to my parents about university | WSU support staff: 1 (100)% | | | |
| I am more motivated to go to university | Year 9: 31 (79.49%)<br>Year 11: 14 (73.68%)<br>Project officers: 3 (100%)<br>WSU support staff: 1 (100)% | Year 9: 6 (15.38%)<br>Year 11: 4 (21.05%) | Year 9: 2 (5.13%)<br>Year 11: 1 (5.26%) | |

### 4.4.1. Year 9

For the year 9 students, the FFP helped increase motivation to go to university, making them feel a sense of welcome and comfort in being on campus, and providing them with a greater understanding of how to manage time more appropriately to cater to their current priorities. As such, the program is preparing students not only for a future in university, but also equipping them with skills that will assist them for the rest of their lives.

### 4.4.2. Year 10

These outcomes reflect a growing internalisation of the aspirations of these young people, as well as the aims of the FFP itself. As the level of self-belief and agency increases for these young people, it is more likely that they will consider university a viable option in their futures.

### 4.4.3. Year 11

Given the short time between their completion of school and the need to move on to either employment or further studies, these outcomes indicate that familiarity, understanding, interest, and motivation have all increased as a result of the program, which encourages the year 11 students to pursue their goals.

### 4.4.4. Year 12

Such positive responses across the various categories in table five underscores the role of the program in addressing needs of year 12 students, increasing their confidence, understanding, interest and familiarity with university life, what is required of them, and using time in a balanced way to assist their present and future aspirations.

### 4.4.5. Parents

A deeper awareness of future educational opportunities and the services afforded by universities has been delivered by the program, and has enabled students to feel more comfortable on campus and improve their time management skills.

### 4.4.6. Project Officers

Due to their belief in the program, its goals, and its function in meeting those goals, project officers had a total of nine outcomes, as outlined in Figure 6 below. Project officers reported a marked change in these areas, highlighting the effectiveness of the program to them. Their work involves a broad array of information sharing, aspiration building, relationship and rapport development, and empowerment for students and their teachers, and they are passionate about the work they do and meeting the goals of the program.

### 4.4.7. WSU Support Staff

There was a consensus amongst the staff members that the work they do in the program bolsters understanding of accessing university and the skills needed to be successful (mostly around time management), and also the increased confidence and motivation that students receive as a result of being involved in the program.

### *4.5. Step 3: Understand What Changes*

This step includes describing how change has happened as a result of the FFP. The responses to the above questions have been expressed as theories of change (also known as program logic). Arrows between the first and second columns show connections between the concepts; both of these qualitative concepts are then related to an outcome, correlating the relationships between the qualitative and

quantitative data in a consistent fashion. Each figure represents the change present for each stakeholder group. In the below figures, 'O' is short for outcome.

*4.6. Step 4: Include Only What Is Material*

The combination of the qualitative responses (expressed in the theories of change; Figures 1–7) and the outcomes determined what is most valuable to each stakeholder group. Each of the outcomes was then monetized to a financial proxy (Tables 6 and 7) that was considered to have the same or a similar effect for the relevant stakeholder group. For example, feeling more comfortable about going onto a university campus was linked to the price of clothing (for the 'a lot' response) and the price of a mobile phone and monthly credit (for the 'somewhat' response), as money spent on these two items typically makes people feel more comfortable. This logic pervaded all financial proxy selections, which are tabulated below:

**Table 6.** Financial proxies ($AUD) for years 9–12.

| | Item | Cost Per Week | Calculation | Cost Per Year |
|---|---|---|---|---|
| 1. | Personal space | $150 per week | $150 × 52 | $7800 |
| 2. | Work | $6.21–$11.52 per hour | 12 h per week. Round up to $80–$140 a week, exact = 12 × $6.21–$11.52 = $74.52–$138.24 × 52 | $4160–$7280 |
| 3. | Food | $50 per week | $50 × 52 | $2600 |
| 4. | Clothing | $50 per week | $50 × 52 | $2600 |
| 5. | Mobile | $800 outright purchase of phone; $50 credit per month | ($50 × 12) + 800 | $1400 |
| 6. | Internet (social media) | $100 per month | $100 × 12 | $1200 |
| 7. | Gaming | $500 outright purchase of console; $10 per week | ($10 × 52) + 500 | $1200 |
| 8. | Transport | $15–$30 per week | $15 − $30 × 52 | $780–$1560 |

**Table 7.** Financial proxies ($AUD) for parents/project officers/X support staff.

| | Item | Cost Per Week | Calculation | Cost Per Year |
|---|---|---|---|---|
| 1. | Work | $16.87–$46.21 per hour | 35–40 h per week. $16.87–$63.40 × 35–40 = $674.80–$2219 × 52 | $35,089.60–$115,388 |
| 2. | Mortgage/rent | $600 per week | $600 × 52 | $31,200 |
| 3. | Food | $150 per week | $150 × 52 | $7800 |
| 4. | Transport/vehicle maintenance | $150 per week | $150 × 52 | $7800 |
| 5. | Bills | $100 per week | $100 × 52 | $5200 |

**Table 7.** Cont.

| | Item | Cost Per Week | Calculation | Cost Per Year |
|---|---|---|---|---|
| 6. | Leisure income | $100 per week | $100 × 52 | $5200 |
| 7. | Clothing | $50 per week | $50 × 52 | $2600 |
| 8. | Mobile | $800 outright purchase; $100 credit per month | ($100 × 12) + $800 | $2000 |
| 9. | Internet | $100 per month | $100 × 12 | $1200 |

### 4.7. Step 5: Do Not Overclaim

This analysis claims only what Fast Forward is responsible for. Each of the outcomes that were realized by stakeholder groups were given a certain amount of deadweight (what would happen without the program), and attribution (who else is contributing to the change), both measured as percentages. It was decided that displacement does not take place within the FFP, as the students that take part in it are selected by the schools and do not take part in other universities' widening participation programs, as per the understanding that the Widening Participation management team have with each school. Drop off was not measured in this analysis, as the FFP's effects were considered to last only for the year that the program takes place, as the subsequent years of involvement therein (from years 9 to 12 for students) builds on the previous year.

### 4.8. Step 6: Be Transparent

In seeking to complete this analysis on the FFP, the Office of Widening Participation sought to evaluate its effectiveness within its communities. As such, there is an inherent belief amongst the authors and the WSU staff that were involved as participants that the FFP is effective and useful in the schools that engage with it. Whilst this is the case, as a matter of course in completing the surveys, participants were encouraged to highlight both the positive and negative experiences of taking part in the program.

The figures presented in this report were entered into a specially designed Excel spreadsheet (Available from http://www.socialvalueuk.org/resource/blank-value-map/) that calculated the overall cost-to-benefit (also referred to as the SROI) ratio, similar to a cost-benefit analysis. The SROI ratio compared the investment with the financial, social, and environmental returns of the program. Taken together, the dollar-for-dollar cost of the program was determined as 1:5.73, or $5.73 (AUD)for every $1 spent, reflecting a very financially viable program. This figure reflects not only the fiscal value of the program, but the important social and aspirational impact of the FFP. Thousands of students for more than a decade have been exposed to university life, had their aspirations bolstered, and have come to realize that despite their postcode or past experiences, university and higher education in general is not beyond their reach, and the dreams and goals that so many students and their parents have for them can indeed be realized through such programs across the university sector. Figure 8 shows excerpts of the impact map, as well as how the different elements combined to form this final cost-to-benefit ratio. More information on the details of the value map are available via the social value URL provided.

| | Stakeholders | Intended/unintended changes | Inputs | Outputs | The Outcomes (what changes) | | | | | | | | | Deadweight % | Displacement % |
|---|---|---|---|---|---|---|---|---|---|---|---|---|---|---|---|
| | | | | | Description | Indicator | Source | Quantity | Duration | Outcomes start | Financial Proxy | Value in currency | Source | | |
| | Who do we have an affect on? Who has an effect on us? | What do you think will change for them? | What do they invest? | What is the value of the inputs in currency *(only enter numbers)* | Summary of activity in numbers | How would the stakeholder describe the changes? | How would you measure it? | Where did you get the information from? | How much change was there? | How long does it last after end of activity? | Does it start in period of activity (1) or in period after (2) | What proxy would you use to value the change? | What is the value of the change? *(Only enter numbers)* | Where did you get the information from? | What would have happened without the activity? | What activity did you displace? |
| | | Better understanding of what University is about | 11 hours (full day + 4 workshops) x $6.21 = $68.31 x 42 (participants) | | 1 full day on campus (9:30am-2:30pm); 2-4 workshops in school (1.5 hour average length). | 1. I am more motivated to go to university | Participants who reported this had changed a lot | Survey | 31 | 1 | 2 | Market value of access to internet | $1,200.00 | $100 per month x 12 months | 15% | 0% |
| | | How to transition from high school to University | | | | | Participants who reported this had changed somewhat | Survey | 6 | 1 | 2 | Market value of access to gaming | $1,020.00 | $500 unit + $10/wk x 12 months | 10% | 0% |
| | | Exposure to University campus and environment | | $2,869.02 | | 2. I am more comfortable about going onto a university campus | Participants who reported this had changed a lot | Survey | 25 | 1 | 2 | Market value of clothing | $2,600.00 | $50 per week x 12 months | 5% | 0% |
| | | Increased awares of how University can benefit future | | | | | Participants who reported this had changed somewhat | Survey | 12 | 1 | 2 | Market value of mobile | $1,400.00 | $800 unit + $50/mth x 12 months | 10% | 0% |
| | | More familiar and motivated to attend University | | | | 3. My understanding of using my time appropriately to cater for homework and study has improved. | Participants who reported this had changed a lot | Survey | 25 | 1 | 2 | Market value of gaming | $1,020.00 | $500 unit + $10/wk x 12 months | 25% | 0% |
| | Yr 9 students | Developed understanding of gaining support to transition to University | | | | | Participants who reported this had changed somewhat | Survey | 11 | 1 | 2 | Market value of transport | $780.00 | $15/wk x 12 months | 15% | 0% |
| | | Cumulative with year 9, including: | 11 hours (full day + 4 workshops) x $7.98 per hour (in kind) = $87.78 x 26 (participants) | | | 1. My understanding that further education and training can help my future has improved. | Participants who reported this had changed a lot | Survey | 22 | 1 | 2 | Market value of mobile | $1,400.00 | $800 unit + $50/mth x 12 months | 30% | 0% |
| | | Awareness of current studies' impact on progression to university / higher education | | | | | Participants who reported this had changed somewhat | Survey | 4 | 1 | 2 | Market value of access to internet | $1,200.00 | $100 per month x 12 months | 25% | 0% |
| | Year 10 students | Increased discernment between differences between TAFE / College and University level study | | $2,282.28 | 1 full day on campus (9:30am-2:30pm); 2-4 workshops in school | 2. My confidence in going onto further study has | Participants who reported this had changed a lot | Survey | 20 | 1 | 2 | Market value of clothing | $2,600.00 | $50 per week x 12 months | 15% | 0% |

**Figure 8.** *Cont.*

| | | | | | | | | | | | | | | | |
|---|---|---|---|---|---|---|---|---|---|---|---|---|---|---|---|
| | | | | | 3. Students' understanding of using their time appropriately to cater for homework and study has improved. | Participants who reported this had changed a lot | Survey | 1 | 1 | 2 | Market value of mobile phone | $2,000.00 | (purchase +) $100 per month x 12 months | 30% | 0% | 30% |
| | | | | | | Participants who reported this had changed somewhat | Survey | 0 | 1 | 2 | Market value of access to internet | $1,200.00 | $100 per month x 12 months | 25% | 0% | 25% |
| University staff providing support in program | Opportunity to engage with high school students | Hourly rate calculated at 115393 per year for lecturer step 4 = 63.40 per hour (7 hour day) x 108 | $6,847.20 | Year 12 Conference (24 staff x 3 hours) = 72 hours; Year 11 day (2 staff x 1.56 hours x 12 days = 36 hours) = 108 | 4. Students are more comfortable about going onto a university campus. | Participants who reported this had changed a lot | Survey | 1 | 1 | 2 | Market value of clothing | $2,600.00 | $50 per week x 52 | 10% | 0% | 10% |
| | | | | | | Participants who reported this had changed somewhat | Survey | 0 | 1 | 2 | Market value of mobile phone | $2,000.00 | ($800 outright purchase +) $100 per month x 12 months | 10% | 0% | 10% |
| | | | | | 5. Students are more familiar with how a university operates. | Participants who reported this had changed a lot | Survey | 1 | 1 | 2 | Market value of mobile phone | $2,000.00 | ($800 outright purchase +) $100 per month x 12 months | 15% | 0% | 15% |
| | | | | | | Participants who reported this had changed somewhat | Survey | 0 | 1 | 2 | Market value of access to internet | $1,200.00 | $100 per month x 12 months | 15% | 0% | 15% |
| | | | | | 6. Students have more confidence to talk to their parents about university. | Participants who reported this had changed a lot | Survey | 1 | 1 | 2 | Market value of mobile phone | $2,000.00 | ($800 outright purchase +) $100 per month x 12 months | 15% | 0% | 10% |
| | | | | | | Participants who reported this had changed somewhat | Survey | 0 | 1 | 2 | Market value of access to internet | $1,200.00 | $100 per month x 12 months | 15% | 0% | 10% |
| | | | | | 7. Students are more motivated to go to university. | Participants who reported this had changed a lot | Survey | 1 | 1 | 2 | Market value of transport / vehicle maintenance | $7,800.00 | $150 per week x 52 weeks | 25% | 0% | 25% |
| | | | | | | Participants who reported this had changed | Survey | 0 | 1 | 2 | Market value of bills | $5,200.00 | $100 per week x 52 weeks | 20% | 0% | 20% |
| **Total** | | | $82,124.11 | | | | | | | | | | | | |

**Figure 8.** *Cont.*

| Stage 4 | → | | | | | Calculating Social Return | | | | | |
|---|---|---|---|---|---|---|---|---|---|---|---|
| Deadweight % | Displacement % | Attribution % | Drop off % | Impact | | | | | | | |
| What would have happened without the activity? | What activity did you displace? | Who else contributed to the change? | Does the outcome drop off in future years? | Quantity times financial proxy, less deadweight, displacement and attribution | | Discount rate | 1.9% | | | | |
| | | | | | | Year 0 | Year 1 | Year 2 | Year 3 | Year 4 | Year 5 |
| 15% | 0% | 5% | 0% | 30,039.00 | | 0.00 | 30,039.00 | 0.00 | 0.00 | 0.00 | 0.00 |
| 10% | 0% | 5% | 0% | 5,232.60 | | 0.00 | 5,232.60 | 0.00 | 0.00 | 0.00 | 0.00 |
| 5% | 0% | 5% | 0% | 58,662.50 | | 0.00 | 58,662.50 | 0.00 | 0.00 | 0.00 | 0.00 |
| 10% | 0% | 5% | 0% | 14,364.00 | | 0.00 | 14,364.00 | 0.00 | 0.00 | 0.00 | 0.00 |
| 25% | 0% | 10% | 0% | 17,212.50 | | 0.00 | 17,212.50 | 0.00 | 0.00 | 0.00 | 0.00 |
| 15% | 0% | 10% | 0% | 6,563.70 | | 0.00 | 6,563.70 | 0.00 | 0.00 | 0.00 | 0.00 |
| 30% | 0% | 20% | 0% | 17,248.00 | | 0.00 | 17,248.00 | 0.00 | 0.00 | 0.00 | 0.00 |
| 25% | 0% | 15% | 0% | 3,060.00 | | 0.00 | 3,060.00 | 0.00 | 0.00 | 0.00 | 0.00 |

**Figure 8.** *Cont.*

| Market value of mobile phone | $2,000.00 | (...... ...... purchase +) $100 per month x 12 months | 10% | 0% | 10% | 0% | 0.00 | | 0.00 | 0.00 | 0.00 | 0.00 | 0.00 | 0.00 |
|---|---|---|---|---|---|---|---|---|---|---|---|---|---|---|
| Market value of mobile phone | $2,000.00 | ($800 outright purchase +) $100 per month x 12 months | 15% | 0% | 15% | 0% | 1,445.00 | | 0.00 | 1,445.00 | 0.00 | 0.00 | 0.00 | 0.00 |
| Market value of access to internet | $1,200.00 | $100 per month x 12 months | 15% | 0% | 15% | 0% | 0.00 | | 0.00 | 0.00 | 0.00 | 0.00 | 0.00 | 0.00 |
| Market value of mobile phone | $2,000.00 | ($800 outright purchase +) $100 per month x 12 months | 15% | 0% | 10% | 0% | 1,530.00 | | 0.00 | 1,530.00 | 0.00 | 0.00 | 0.00 | 0.00 |
| Market value of access to internet | $1,200.00 | $100 per month x 12 months | 15% | 0% | 10% | 0% | 0.00 | | 0.00 | 0.00 | 0.00 | 0.00 | 0.00 | 0.00 |
| Market value of transport / vehicle maintenance | $7,800.00 | $150 per week x 52 weeks | 25% | 0% | 25% | 0% | 4,387.50 | | 0.00 | 4,387.50 | 0.00 | 0.00 | 0.00 | 0.00 |
| Market value of bills | $5,200.00 | $100 per week x 52 weeks | 20% | 0% | 20% | 0% | 0.00 | | 0.00 | 0.00 | 0.00 | 0.00 | 0.00 | 0.00 |
| | | | | | | **Total** | 479,211.40 | | 0.00 | 479,211.40 | 0.00 | 0.00 | 0.00 | 0.00 |
| | | | | | | | | | | 470,276.15 | | | | |
| | | | | | | Present value of each year | | | 0.00 | | 0.00 | 0.00 | 0.00 | 0.00 |
| | | | | | | Total Present Value (PV) | | | | | | | | 470,276.15 |
| | | | | | | Net Present Value | | | | | | | | 388,152.04 |
| | | | | | | (PV minus the investment) | | | | | | | | |
| | | | | | | Social Return | | | | | | | | 5.73 |
| | | | | | | Value per amount invested | | | | | | | | |

**Figure 8.** Excerpts of FFP impact map.

*4.9. Step 7: Verify the Result*

A two-page summary sheet was sent to the schools that participated in the evaluation program, which can be viewed at the Western Sydney University's Widening Participation webpage (Available by contacting the authors). The two-page summary sheet provided further context to the stakeholders, and discussed the key findings and insights of the research. The summary sheet further engages the stakeholders and increases transparency of the SROI process.

## 5. Limitations

One of the advantages of utilizing the SROI framework in this evaluation has been the strength of this methodology to uniquely engage with stakeholders who took part in the program, and who were able to offer varied perspectives as a result of the rapport present between two members of the research team (who worked in the widening participation department of Western Sydney University) and the schools the program engaged with previously. Having this rapport in place to conduct such an evaluation was pertinent to the level of stakeholder response; without it, responses may have been more limited.

Due to the short timeframe of this evaluation (the time from formulation of research to drafting of this article was one year), the research team could have worked more closely with schools to have more in-school meetings to create more specific and nuanced proxies. The time restrictions did not allow for this, as this would be an iterative process that itself would have taken at minimum a few weeks to conduct effectively.

Bias was prevented across all research questions and measures by the research team checking the research questions with widening participation staff at Western Sydney University who were not involved in the evaluation of the programs, but were regularly engaged with the schools, teachers, parents, and students, and were therefore able to offer insight into our questions and their relevance. The figures used for the excel spreadsheet, including the proxies, were also shared with members of the widening participation team to verify that the cost-to-benefit ratio was reasonable and in fact conservative when monetizing the immaterial benefits of the program.

## 6. Conclusions

An SROI methodology allowed for the measuring of impact—both quantitatively and narrative-based. The impact of the FFP shows that it is an important support program for students and parents. The responses from students in the FFP indicated they were better prepared for their future and became more familiar with what higher education has to offer. Students found they had a greater understanding of how further education and training can help their future, how to better use their time to cater for homework and study, were more comfortable about going onto a university campus, and were more motivated to attend university. Similar positive results were found amongst parents who stated they were made more aware of the opportunities further education can give their children and felt more included in the educational journey and thus better able support their child's transition. Project officers and Western Sydney University support staff were encouraged by their participation in the program, and expressed a sense of vitality and purpose to their work. The SROI ratio measures financial value relative to the resources invested. The benefits or social return (present value) accumulated to $470,286.15, with the financial investment (present value) adding up to $82,124.11 for five schools. The gave an SROI ratio of 5.73:1—for every $1 dollar invested, Fast Forward is providing $5.73 in social value. The narrative-based and quantitative data show that the FFP fulfils its objectives of engaging high school communities across greater western Sydney, by preparing students for university and encouraging participation.

**Author Contributions:** Conceptualization, J.R. and S.S.; methodology, J.R., S.S., J.M. and G.P.; formal analysis, J.R., S.S., J.M. and G.P.; investigation, J.R., S.S., J.M. and G.P.; resources, J.R., J.M. and G.P.; data curation, J.R. and S.S.; writing—original draft preparation, J.R. and S.S.; writing—review and editing, J.R., S.S., J.M. and

G.P.; visualization, J.R., S.S., J.M. and G.P.; supervision, J.R., J.M. and G.P.; project administration, S.S.; funding acquisition, J.R., J.M.; G.P.

**Funding:** This research received no external funding.

**Acknowledgments:** The research team would like to acknowledge the cooperation of the Fast Forward Program staff members, who allowed Western Sydney University to visit their in-school sessions. Their assistance and guidance throughout this project was crucial for its realization. Thank you.

**Conflicts of Interest:** The authors declare no conflict of interest. The funders had no role in the design of the study; in the collection, analyses, or interpretation of data; in the writing of the manuscript, or in the decision to publish the results.

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
