# Peer review of "Utilising the Social Return on Investment (SROI) Framework to Gauge Social Value in the Fast Forward Program"

_education, doi:10.3390/educsci9040290_

Round 1

Reviewer 1 Report

This paper addresses and provides a solution for the ongoing issue of demonstrating social value and impact of efforts to widen access and participation in Australia. The authors are to be congratulated for identifying a methodology that can be used across the sector and for the applied example that demonstrates how to implement and report on SROI.

To ensure that practitioners and researchers in widening participation can fully utilise the SROI, it would be advantageous to elaborate in Section 3 Methodology all of the steps involved e.g. via a model/visual or infographic. Furthermore, please share with the readers more explicitly that this project applied a methodology from Social Value International as this will be of great value to practitioners.  Please note that the hyperlink provided in footnote 3 is not working/correct and needs to be updated. 

Similarly, it would be beneficial to practitioners and researchers if some aspects of the completed Excel spreadsheet that calculates the final $5.73 SROI is provided as evidence. The reader, in effect, has to take the word of the authors in section 4.7 and perhaps screenshots or details in a Technical Appendix of the actual calculation is needed. In time, this spreadsheet may change or be removed so it is important that the calculation is captured in some way to verify the paper's findings.

Overall, a well-written paper that presents a useful approach for all involved in widening participation efforts.

Author Response

To ensure that practitioners and researchers in widening participation can fully utilise the SROI, it would be advantageous to elaborate in Section 3 Methodology all of the steps involved e.g. via a model/visual or infographic. Furthermore, please share with the readers more explicitly that this project applied a methodology from Social Value International as this will be of great value to practitioners.  Please note that the hyperlink provided in footnote 3 is not working/correct and needs to be updated.  

All sections have now been elaborated (see track changes throughout). The article now references Social Ventures Australia (see pp. 4-5). The URL to the value map has now been updated (p.32).  

Similarly, it would be beneficial to practitioners and researchers if some aspects of the completed Excel spreadsheet that calculates the final $5.73 SROI is provided as evidence. The reader, in effect, has to take the word of the authors in section 4.7 and perhaps screenshots or details in a Technical Appendix of the actual calculation is needed. In time, this spreadsheet may change or be removed so it is important that the calculation is captured in some way to verify the paper's findings. 

Excerpts of the spreadsheet have now been included as part of the main text (pp.33-36).  

Reviewer 2 Report

Reviewer Report on “Utilising the Social Return on Investment (SROI) framework to gauge social value in the Fast Forward Program”

General comments. This paper under review applies a SROI (Social Return on Investment) method to evaluate a program (the Fast Forward Program) run by a university in Australia. The paper provides a potentially useful case study for understanding possible approaches to promoting higher education participation in Australia. However, the paper is ill-structured. In particular, the Method section is far from adequate and clear; descriptions of key components of the method employed are not introduced until in the Findings section. The Findings section should be made more organized – the current form looks like a lab report. Also, many key terms and arguments in the paper are vague and sometimes rather confusing. Thus, the paper should be carefully and thoroughly revised to have a better and clearer structure; more information should also be provided so that the paper can better convey useful information to the reader.

Comments on specific sections.

On background information. Some background information should be added so that arguments in the paper can be understood more easily. For example, it may be helpful to provide some background information on the (higher) education system in Australia, especially on how high school students make university choices and how schools select applicants. For example, do students need to take college entrance exams to compete for a spot in a college, as in the case of China? Or do they only need to submit an essay application, as in the case of the U.S.? Without information on the context of the higher education system, it is hard for the reader to appreciate the value of the FFP program.

On the literature. The literature review is inadequate. For example, the LR fails to answer: are there other, similar programs being run in Australia and elsewhere? How were they evaluated (i.e. methods? Data? Findings)? How were previous studies helpful in guiding the present study in choosing the most suitable evaluation method? And so on.

On methodology. The Method section is inadequate. First, apart from its name, what type of study is being conducted is not clear. The paper should explicit state upfront: What type of study is being performed? For example, qualitative or quantitative? If it’s a qualitative study, what specific methods were used, e.g. focus group, participatory evaluation, or something else? If it’s quantitative, then what kind of models are being estimated? And so on.

Second, some key components of the method adopted are missing. The Method section begins by talking about the goals of the SROI method and mentioning how it was “described” and “considered”, with talking about what it is. Also, what are the main advantages of SROI over other evaluation methods (so that SROI is chosen over the other alternatives)? Detailed steps of applying this method are not introduced until the Findings section, when the actual evaluation exercises are being discussed, which makes these steps rather confusing (as they are mixed up with the survey questions and the findings). In fact, I think a one-or-two-paragraph description of the 7 steps, perhaps along with a diagram depicting the key components of the method and the key steps of carrying out this method, should be added to make the Method section easier to understand.

Third, what are the survey instruments? The paper should provide some examples of the survey questions in the Method section, rather than to abruptly bring them up in the Findings section.

Fourth, it is never clear how the 109 participants were selected. How representative are these participants of the underly population? Are they non-responses?

Finally, there is no mention of how “social returns” are measured anywhere in the entire Method section!!!

On the findings. Many of the findings discussed in the paper are not related to the dollar value of the social impact of FFP. So, either the paper should be shortened to focus more on the value of the social impact, or the title and the introduction should be changed to include more aspects of the program. On limitations. The paper does not discuss any limitations of the study or of the FFP program. How likely can the findings of a program in a single university be generalized to a broader context? Does it always work? When and where will it perform better? Is it possible that something done when carrying out the 7 steps lead to biased findings? What measures were taken to prevent biased findings?

More specific comments.

Line 10. What’s X? Here it seems to be a university, but elsewhere (e.g. line 31) it seems to be a region or a district in Australia. Please clarify. This comment applies to all the X’s appearing in the paper.

Line 28. What do you mean by “non-traditional” students? International readers not familiar with the Australian context might find it difficult to understand. Please clarify.

Line 38. Please define “social impact” here.

Lines 40-42. Any references of this methodology (SROI)? Or, is this methodology originally devised in this paper? May need to explain a bit more (than “narrative” and “quantitative”) here on: what is SROI? How does it work? And, why is it useful (compared to other possible methods)?

Line 43. Before jumping into the literature review, it may be helpful to provide some background information about the education system in Australia, especially on how high school students transition into colleges and universities. For example, do they need to take college entrance exams to compete for a spot in a college? Are tuitions so high that they prevent some groups of students from entering colleges? Otherwise, some materials in the literature may not be easy to understand. For example, why “social justice” would be an issue and why students’ socio-economic backgrounds matter that much even in a rich country like Australia?

Related to these comments, in lines 61-63, the paper says that “Students who come from low socio-economic backgrounds…. with parents… that are not as familiar with the higher education system”. Yet why might that (being unfamiliar with the higher education system) be? In particular, what features of the higher education system should these students and their parents know about?

Also related to these comments, what “academic culture (line 74)” should be demystified?

Line 48. Again, what does “non-traditional backgrounds” mean?

Line 90. Again, what does “non-traditional students” mean?

Lines 122-123. IRB approval number? How consents were provided by the participants? And what is NSW?

Table 1: What do “Year 9” and “Year 12” mean? Are these “years since program implementation”? Or the ages of children (If so, why not call them “age 9” and “age 12”)? And how about Year 10 and Year 11? Not in the Table? The structure of Table 1 is rather confusing (parents from all years did the survey but not all participated in the events described in the Table?). The author should present the information in a clearer manner.

Lines 139-141 should be put right below line 131.

Lines 143-170. Since there are a lot in common among different “Years”, why not summarize them in a table (or some chart) and talk about the key differences?

Line 172. This sentence should be moved to somewhere close to line 132.

Line 184. The “theory of change” figures should be moved to the Method section—they are part of your method. Why not use them to guide and summarize your findings? It is really not a good way to list and discuss the questions and responses one by one in a lab-report way. You should summarize them. What are the take-home messages?

The rest of the paper should be re-organized in a more structured and coherent way, rather than discussing the survey questions and the findings in an item-by-item manner.

Author Response

Reviewer 2

General comments. This paper under review applies a SROI (Social Return on Investment) method to evaluate a program (the Fast Forward Program) run by a university in Australia. The paper provides a potentially useful case study for understanding possible approaches to promoting higher education participation in Australia. However, the paper is ill-structured. In particular, the Method section is far from adequate and clear; descriptions of key components of the method employed are not introduced until in the Findings section. The Findings section should be made more organized – the current form looks like a lab report. Also, many key terms and arguments in the paper are vague and sometimes rather confusing. Thus, the paper should be carefully and thoroughly revised to have a better and clearer structure; more information should also be provided so that the paper can better convey useful information to the reader.

These general comments have been applied to the article as a whole.

Note that the theory of change has been placed specifically where it has, as this conforms to the chronological methodological sequence of how the SROI methodology is applied. In this way, the authors wish to maintain the current structure, as it creates a template for other research / evaluation papers to utilise the SROI framework in its present form. The cumulative impact of showing each stakeholder group’s responses to the individual questions, item by item, shows the different changes that have occurred in a systematic way.

It is argued that the theories of change are not part of the method, but are derived as a result of the methodology being conducted – they form a summarised version of what was found as a part of the research.

The article has been edited and updated to include more information so that the article is clearer and more useful information has been provided.

On background information. Some background information should be added so that arguments in the paper can be understood more easily. For example, it may be helpful to provide some background information on the (higher) education system in Australia, especially on how high school students make university choices and how schools select applicants. For example, do students need to take college entrance exams to compete for a spot in a college, as in the case of China? Or do they only need to submit an essay application, as in the case of the U.S.? Without information on the context of the higher education system, it is hard for the reader to appreciate the value of the FFP program.

(See p. 2)

The Australian Higher Education system allows domestic students to apply to the university of their choice once they have completed their Higher School Certificate or equivalent – different states have different names for the same qualification (for example, the Victorian Certificate of Education). Students’ scores are determined by their Australian Tertiary Admissions Ranking or ATAR, with specific courses having a specific ATAR score required for entry. ATARs are determined by a combination of in-class assessments completed throughout the final year of high school study, and performance in a range of examinations throughout the year. Students apply beginning in August for the courses they wish to study via the University Admissions Centre or UAC (see https://www.uac.edu.au/future-applicants/how-to-apply-for-uni for more information), and they are offered a place at the university of their choice based on how they have performed against other students who have also applied for the course. Some courses have extra or other entrance requirements, such as medicine, which are determined by each university. Other entry pathways are also possible, such as applying directly to the university (https://www.uac.edu.au/future-applicants/how-to-apply-for-uni) or studying at Technical And Further Education centres (see https://www.tafensw.edu.au/tafe-nsw for more information) or other colleges that offer pre-university level certificates, that can be used to transition into university courses. Tuition fees are typically offset by the HECS-HELP scheme, where domestic students take out a loan from the Government which is paid back upon earning a particular income threshold, deducted from one’s payslips at each pay cycle (https://www.studyassist.gov.au/help-loans/hecs-help). As such, finances do not typically hinder students from attending university.

On the literature. The literature review is inadequate. For example, the LR fails to answer: are there other, similar programs being run in Australia and elsewhere? How were they evaluated (i.e. methods? Data? Findings)? How were previous studies helpful in guiding the present study in choosing the most suitable evaluation method? And so on.

(See pp.3-4).

Factors that assist school students aged 14 to 19 years to engage meaningfully with the learning process have also been explored by scholars [18]. It was found that disengagement in school occurs as a result of three overriding factors: the depth of relationships shared peer to peer and teacher to peer; the quality of the teaching, and the perceived relevance of more “practical / vocational subjects compared with academic subjects” [18, p.113]. These factors lead to “[a] defined relatedness with respect to school climate, teacher relationships, feelings of belonging and acceptance and inter-personal support” [18, p.113], all of which contribute to how students engage with their studies.

2.4 Comparative evaluations of widening participation programs

Some of the evaluations that have taken place in Australia include the National Centre for Student Equity in Higher Education or NCSEHE’s evaluation of 31 widening participation initiatives around the country [19]. These programs and support services are funded in Australia by the Higher Education Participation and Partnerships Program (HEPPP) funding scheme. The NCSEHE reported on the importance of university partnerships, and how successful outcomes where ascertained throughout these projects. Some of the goals of these projects include offering support to students from non-tradition backgrounds, interventions to assist accessing university courses via bridging and other programs, and encouraging them to maintain and complete their studies [19].

Widening participation programs at one Australian university target non-traditional students, and have developed perspectives towards more inclusive practices within widening participation [20, para.1]. Other scholars identified key components that increase the effectiveness of widening participation programs, namely meaningful collaboration between communities and stakeholders and university staff [21, p.40]; on-campus experiences, such as information sessions and campus tours [21, p. 41]; mentoring from university students and academics, who act as role models [21, p.41], and collaborative learning at school, where widening participation staff engage with schools to develop and deliver relevant and aspiration-building content taught in schools [21, p.43]. In terms of evaluation, students’ confidence to come to university, study a particular course of study (e.g. science), and increased knowledge around alternative means of university entry were increased, amongst others [21, pp.37-39].

Similarly in the U.K. and U.S., widening participation programs target students from non-traditional backgrounds [22]. Evaluations here are conducted by using the “counterfactual condition” – realising the impact and changes produced by the program by comparing it to similar settings without a program [22, p.744]. Randomization and quasi experimental designs are also given as effective evaluative models [22].

Widening participation is the “key mechanism” where universities can reach non-traditional students and promote university attendance, however, evaluation is generally scarce and “little is known about their effectiveness” [23, p.1384]. There is a need for a greater commitment to evaluating widening participation programs [24]. Evaluation of widening participation programs is critically important to “facilitate both programme development and the creation of an evidence base guide to future practice and policy” [24, p.384]. This article addresses this imperative in the literature by considering how the FFP functions to increase aspirations for the students it works with, and how the program fosters these aspirations alongside relationships between the different stakeholder groups, which collectively assist in highlighting university as a viable option for the students that engage with the program.

On methodology. The Method section is inadequate. First, apart from its name, what type of study is being conducted is not clear. The paper should explicit state upfront: What type of study is being performed? For example, qualitative or quantitative? If it’s a qualitative study, what specific methods were used, e.g. focus group, participatory evaluation, or something else? If it’s quantitative, then what kind of models are being estimated? And so on.

(See pp.4-5).

Social Ventures Australia (Social Ventures Australia, 2018) developed the SROI methodology, which is currently being used by a range of organizations to assess the social impact of their programs. This methodology was chosen as a relevant and meaningful tool as it measures impact at multiple levels when compared to typical cost-benefit analyses, and features a strong narrative element in the incorporation of the assessment of social impact and change. Stakeholders and not researchers determine those elements of the program that are considered most valuable, and the programs are therefore evaluated for the social purposes they are serving (Nicholls et al. 2012). Both qualitative and quantitative measures contribute towards the development of a cost to benefit ratio that monetizes immaterial social changes that took part due to the FFP taking place, such as increased aspiration to attend university and increased confidence on campus. SRIO’s ability to monetize these attributes is unique, and provides a more holistic account of how change happened within the communities that engaged with the FFP. The strong narrative focus also allows for a fuller understanding of how the methodology captures change, and was therefore considered the most relevant methodological tool for the current research project.

Second, some key components of the method adopted are missing. The Method section begins by talking about the goals of the SROI method and mentioning how it was “described” and “considered”, with talking about what it is. Also, what are the main advantages of SROI over other evaluation methods (so that SROI is chosen over the other alternatives)? Detailed steps of applying this method are not introduced until the Findings section, when the actual evaluation exercises are being discussed, which makes these steps rather confusing (as they are mixed up with the survey questions and the findings). In fact, I think a one-or-two-paragraph description of the 7 steps, perhaps along with a diagram depicting the key components of the method and the key steps of carrying out this method, should be added to make the Method section easier to understand.

(See pp.5-6).

1. Involve stakeholders. Stakeholders (i.e. participants) inform what is considered meaningful to the Fast Forward Program within their context and how the program is measured and valued;

2. Understand what changes. Describing how change has happened as a result of the Fast Forward Program. This change is not limited to positive and intended change, but also includes negative and unintended change. These changes are understood as outcomes, which form a theory of change – how Fast Forward has impacted the community, and how these changes are measured for all participants. Each outcome is measured through indicators, validating their effectiveness.

3. Value the things that matter. Outcomes such as increased aspiration and motivation are given financial proxies in order to value how the outcomes can be measured quantitatively.

4. Only include what is material. Deciding what information needs to be included when assessing the program, so that an accurate picture of the program’s impact can be ascertained. This step uses the term ‘materiality’, where “information is material if missing it out of the SROI would misrepresent the [program’s] activities” (Nicholls et al. 2012, p.9).

5. Do not over claim. This analysis claims only what Fast Forward is responsible for. An example of overclaiming would be attributing an increase in motivation to the program when in fact students’ parents have continuously encouraged university participation throughout their schooling careers. Deadweight refers to the amount, expressed as a percentage, of the outcomes that would have happened without Fast Forward being present in the school. Attribution is similar to deadweight, except that it measures if / how these outcomes would have been met by other people (students, parents, teachers, other influences). The final consideration in this section is drop-off, which measures how the impact of the outcomes depreciate over time, especially the outcomes that are derived solely from the Fast Forward Program. It may be that, over time, other factors or influences impact the participants more than the program itself. Drop-off is considered for outcomes that last for more than one year; this analysis considers the impact of the program for only one year (2017), so drop-off was not applied in this analysis.

6. Be transparent. Reflect the honest perspectives of research participants, as well as the authors’ position as those who want to show Fast Forward to be an effective program. At this point, theory of change is developed, showing the processes of change that have occurred, and how this has led to outcomes being met. The SROI ratio of money invested: money gained, based on the commodification of the outcomes as material, is also developed at this point, which highlights the economic impact of the program whilst drawing upon the social, educational and other outcomes that have been delivered as a result of the program.

7. Verify the result. Present findings to all stakeholders via emailing a 2 page report to school representatives, which was distributed to those that took part in the research.

Third, what are the survey instruments? The paper should provide some examples of the survey questions in the Method section, rather than to abruptly bring them up in the Findings section.

(See p. 7, with accompanying table 2)

Participants were asked to complete a paper (for students) and online (for all other groups) survey that asked questions around their personal background (year group, gender, school attended / length of time in currently employed position, relationship to university (did you / your parents attend university, do you / your parents encourage you to go to university), as well as their perceptions of the best parts of the program (qualitative: what are the best parts of the program, what have you learned from being involved?). The remaining three questions asked about the value of completing school and balancing time, alongside responses to the changes and improvements that occurred as a result of the program, to be discussed below (see value the things that matter, pp 14-24).

Fourth, it is never clear how the 109 participants were selected. How representative are these participants of the underly population? Are they non-responses?

(see p. 7)

The research team gave the selected schools consent forms and information sheets, which the schools in turn gave the year 9-12 students to take home to obtain parental consent to take part in the evaluation. Those students who returned a signed consent form were able to complete the paper survey. One of the research team members negotiated a time and date to visit the school, with teachers from the schools organising for students to be arranged in particular classrooms so that they could complete the evaluation. The research team member met the students, provided a verbal explanation of the research and what they were being asked to do, and completed the survey. All other stakeholder groups were emailed an invitation to take part in the research, and this email contained a URL with a link to the survey to complete.

Finally, there is no mention of how “social returns” are measured anywhere in the entire Method section!!!

The social returns are not part of the method sections in the minds of the authors. The method involves data collection and analysis; the presentation of the findings, including the social returns and impact of the program, are presented in the findings as this is what was found as a result of the research.

On the findings. Many of the findings discussed in the paper are not related to the dollar value of the social impact of FFP. So, either the paper should be shortened to focus more on the value of the social impact, or the title and the introduction should be changed to include more aspects of the program.

SROI as a methodology is not solely focussed on the dollar value. The methodology has been expounded (pp.4-9), to show that the focus of this evaluative tool is not only looking at the dollar for dollar value, but the social impact of the program overall, which includes narrative approaches, which have been expounded throughout (see Findings section, pp. 10-37).

On limitations. The paper does not discuss any limitations of the study or of the FFP program. How likely can the findings of a program in a single university be generalized to a broader context? Does it always work? When and where will it perform better? Is it possible that something done when carrying out the 7 steps lead to biased findings? What measures were taken to prevent biased findings?

(see p.37)

One of the advantages of utilizing the SROI framework in this evaluation has been the strength of this methodology to uniquely engage with stakeholders that took part in the program, who were able to offer varied perspectives as a result of the rapport that was existent between two members of the research team (who worked in the widening participation department of X) and the schools the program engaged with previously. Having this rapport in place in order to conduct such an evaluation was pertinent to the level of response; without this rapport, responses may have been more limited. Due to the short timeframe of this evaluation (the time from formulation of research to drafting of the article was one year), the research team could have worked more closely with schools to have more in-school meetings to create more specific and nuanced proxies. The time restrictions did not allow for this, as this would be an iterative process that itself would have taken at minimum a few weeks to conduct effectively. Bias was prevented across all research questions and measures by the research team checking the research questions with widening participation staff at X university who were not involved in the evaluations of the programs, but were regularly engaged with the schools, teachers, parents and students, and were therefore able to offer insight into our questions and their relevance. The figures used for the excel spreadsheet, including the proxies, were also shared with members of the widening participation team, to verify that the cost to benefit ratio was reasonable and in fact conservative when monetizing the immaterial benefits of the program. 

Line 10. What’s X? Here it seems to be a university, but elsewhere (e.g. line 31) it seems to be a region or a district in Australia. Please clarify. This comment applies to all the X’s appearing in the paper.

The symbol X has been used to ensure the anonymity of the reference ‘Western Sydney’ or ‘Western Sydney University’. The reason there are two separate uses (one the region and the other the university) is so that readers cannot deduce from the regional reference which university is being spoken of.

Line 28. What do you mean by “non-traditional” students? International readers not familiar with the Australian context might find it difficult to understand. Please clarify.

Line 32-33 Within this context, non-traditional students are those who are typically the first in their family to attend university, from migrant and / or from non-English speaking backgrounds.

Line 38. Please define “social impact” here.

Line 40-41: That is, how perceptions and aspirations of attending university have changed

Lines 40-42. Any references of this methodology (SROI)? Or, is this methodology originally devised in this paper? May need to explain a bit more (than “narrative” and “quantitative”) here on: what is SROI? How does it work? And, why is it useful (compared to other possible methods)?

See methodology pp.4-9.

Line 43. Before jumping into the literature review, it may be helpful to provide some background information about the education system in Australia, especially on how high school students transition into colleges and universities. For example, do they need to take college entrance exams to compete for a spot in a college? Are tuitions so high that they prevent some groups of students from entering colleges? Otherwise, some materials in the literature may not be easy to understand. For example, why “social justice” would be an issue and why students’ socio-economic backgrounds matter that much even in a rich country like Australia?

(See p.2)

The Australian Higher Education system allows domestic students to apply to the university of their choice once they have completed their Higher School Certificate or equivalent – different states have different names for the same qualification (for example, the Victorian Certificate of Education). Students’ scores are determined by their Australian Tertiary Admissions Ranking or ATAR, with specific courses having a specific ATAR score required for entry. ATARs are determined by a combination of in-class assessments completed throughout the final year of high school study, and performance in a range of examinations throughout the year. Students apply beginning in August for the courses they wish to study via the University Admissions Centre or UAC (see https://www.uac.edu.au/future-applicants/how-to-apply-for-uni for more information), and they are offered a place at the university of their choice based on how they have performed against other students who have also applied for the course. Some courses have extra or other entrance requirements, such as medicine, which are determined by each university. Other entry pathways are also possible, such as applying directly to the university (https://www.uac.edu.au/future-applicants/how-to-apply-for-uni) or studying at Technical And Further Education centres (see https://www.tafensw.edu.au/tafe-nsw for more information) or other colleges that offer pre-university level certificates, that can be used to transition into university courses. Tuition fees are typically offset by the HECS-HELP scheme, where domestic students take out a loan from the Government which is paid back upon earning a particular income threshold, deducted from one’s payslips at each pay cycle (https://www.studyassist.gov.au/help-loans/hecs-help). As such, finances do not typically hinder students from attending university.

Line 48. Again, what does “non-traditional backgrounds” mean?

See lines 32-33.

Related to these comments, in lines 61-63, the paper says that “Students who come from low socio-economic backgrounds…. with parents… that are not as familiar with the higher education system”. Yet why might that (being unfamiliar with the higher education system) be? In particular, what features of the higher education system should these students and their parents know about?

Also related to these comments, what “academic culture (line 74)” should be demystified?

Lines 94-100: This lack of understanding of how the university system works, particularly how transition from high school to university takes place and the kinds of social and academic supports their children may need, does little to aid students’ ability to consider higher education as a viable option for their future. Parents may not be aware of these realities due to their non-attendance at university themselves, especially if they came from lower socio-economic status

Clarified: which can be a barrier to aspiring towards higher educational attainment due to a lack of understand how it works

Line 90. Again, what does “non-traditional students” mean?

See lines 32-33.

Lines 122-123. IRB approval number? How consents were provided by the participants? And what is NSW?

Line 264: X ethics number H12079. 

Lines 294-303: The research team gave the selected schools consent forms and information sheets, which the schools in turn gave the year 9-12 students to take home to obtain parental consent to take part in the evaluation. Those students who returned a signed consent form were able to complete the paper survey. One of the research team members negotiated a time and date to visit the school, with teachers from the schools organising for students to be arranged in particular classrooms so that they could complete the evaluation. The research team member met the students, provided a verbal explanation of the research and what they were being asked to do, and completed the survey. All other stakeholder groups were emailed an invitation to take part in the research, and this email contained a URL with a link to the survey to complete. Schools contacted parents whose children took part in the FFP with this same email with the survey to invite them to take part. 

Table 1: What do “Year 9” and “Year 12” mean? Are these “years since program implementation”? Or the ages of children (If so, why not call them “age 9” and “age 12”)? And how about Year 10 and Year 11? Not in the Table? The structure of Table 1 is rather confusing (parents from all years did the survey but not all participated in the events described in the Table?). The author should present the information in a clearer manner.

Lines 272 – 274: The term ‘Year 9’ or other number means the year of education attained. In Australia, high school commences at Year 7 (11-13 years of age) and finishes at Year 12 (16-18 years of age).

Line 278: The table has been adjusted to reflect that Years 9 – 11 took part in the same kinds of activities, apart from the Year 9 Welcome to X Evening, which is mentioned separately in the table.

Lines 139-141 should be put right below line 131.

The content has been moved – see lines 274-276.  

Lines 143-170. Since there are a lot in common among different “Years”, why not summarize them in a table (or some chart) and talk about the key differences?

There are some similarities, but there are at some points slight and other points larger differences. The cumulative impact of each year shows the unique social impact for each year group, and has therefore been retained.

Line 172. This sentence should be moved to somewhere close to line 132.

In order to attach the comment of only females taking part and connecting this to the influence of mothers, this sentence has not been moved. As it stands, this line is the first analysis of any parental views, and as such it is more important (and less to remember) for the reader, so that the connection between female participation and mothers’ influence on educational aspiration is clear.

Line 184. The “theory of change” figures should be moved to the Method section—they are part of your method. Why not use them to guide and summarize your findings? It is really not a good way to list and discuss the questions and responses one by one in a lab-report way. You should summarize them. What are the take-home messages?

The theories of change function as a summary of the qualitative data presented for the two qualitative questions asked of participants. The take home messages are the outcomes for each stakeholder group, as listed in the outcomes column of each theory of change map. The qualitative material expressed above give detailed explanations of the findings. A key element of the SROI approach is to include a significant amount of narrative from stakeholders, which is why more detailed and then summarised (theory of change) maps are presented.

Round 2

Reviewer 2 Report

The manuscript has been greatly improved; most of the "missing pieces" have been added to the paper. Yet it still seems that it's too lengthy and not concise and focused enough (especially in the "results" section). I think the "results" section should be tightened to further improve the quality of communication.

Author Response

Pages 10-21 of the attached document have been reviewed and made more concise.